# CAR T-cells targeting FGFR4 and CD276 simultaneously show potent antitumor effect against childhood rhabdomyosarcoma

Meijie Tian [1,5], Jun S. Wei [1,5], Adam Tai-Chi Cheuk[1], David Milewski [1], Zhongmei Zhang[1], Yong Yean Kim [1], Hsien-Chao Chou [1], Can Liu [2], Sherif Badr[3], Eleanor G. Pope [1], Abdelrahman Rahmy[1], Jerry T. Wu[1], Michael C. Kelly [4], Xinyu Wen[1] & Javed Khan [1] ✉

Chimeric antigen receptor (CAR) T-cells targeting Fibroblast Growth Factor Receptor 4 (FGFR4), a highly expressed surface tyrosine receptor in rhabdomyosarcoma (RMS), are already in the clinical phase of development, but tumour heterogeneity and suboptimal activation might hamper their potency. Here we report an optimization strategy of the co-stimulatory and targeting properties of a FGFR4 CAR. We replace the CD8 hinge and transmembrane domain and the 4-1BB co-stimulatory domain with those of CD28. The resulting CARs display enhanced anti-tumor activity in several RMS xenograft models except for an aggressive tumour cell line, RMS559. By searching for a direct target of the RMS core-regulatory transcription factor MYOD1, we identify another surface protein, CD276, as a potential target. Bicistronic CARs (BiCisCAR) targeting both FGFR4 and CD276, containing two distinct co-stimulatory domains, have superior prolonged persistent and invigorated anti-tumor activities compared to the optimized FGFR4-specific CAR and the other BiCisCAR with the same 4-1BB co-stimulatory domain. Our study thus lays down the proof-of-principle for a CAR T-cell therapy targeting both FGFR4 and CD276 in RMS.

Chimeric antigen receptor (CAR) T-cell therapies targeting cancer-specific antigens have impressive successes in treating refractory and relapsed leukemia and lymphoma[1,2]. However, they have thus far displayed poor efficacy in treating solid tumors due to several challenges, including heterogenous expression of tumor-associated antigens, loss of target expression under selection pressure, limited T-cell potency, inadequate trafficking, a hostile tumor microenvironment, the propensity of exhaustion and lack of persistence of the CAR T cells[3–5]. Overcoming these hurdles for the treatment of solid tumors remains a significant challenge and an area of active investigation.

Rhabdomyosarcoma (RMS) is the most common soft tissue sarcoma in children and represents approximately 3–4% of all childhood cancers[6]. We previously reported high expression of Fibroblast Growth Factor Receptor 4 (FGFR4) in both PAX3/7-FOXO1 fusion-positive (FP-) and fusion-negative (FN-) RMS, but low or absent in normal tissues[7,8]. Furthermore, *FGFR4* is a direct transcriptional target of fusion protein PAX3-FOXO1[9,10], driving high expression in FP-RMS. Additionally, ~10 % of FN-RMS have activating mutations with high expression of FGFR4[11–13]. These characteristics make FGFR4 a tractable molecular target for RMS, including CAR T therapy.

[1]Genetics Branch, Center for Cancer Research, National Cancer Institute, National Institutes of Health, Bethesda, MD 20892, USA. [2]Multiscale Systems Biology Section, Laboratory of Immune System Biology, NIAID, NIH, Bethesda, MD 20892, USA. [3]Experimental Immunology Branch, Center for Cancer Research, National Cancer Institute, National Institutes of Health, Bethesda, MD 20892, USA. [4]Single Cell Analysis Facility, Center for Cancer Research, National Cancer Institute, National Institutes of Health, Bethesda, MD 20892, USA. [5]These authors contributed equally: Meijie Tian, Jun S. Wei. ✉e-mail: khanjav@mail.nih.gov

CD276 (B7-H3) is another cell surface protein belonging to important immune checkpoint B7 families[14]. We and others have reported that CD276 is overexpressed on a wide range of human solid tumors, including RMS[15–17], and its overexpression is correlated with tumor progression, metastasis, and poor clinical outcome across a variety of malignancies[18,19]. Furthermore, PAX3-FOXO1 up-regulates *CD276* expression in FP-RMS[20] and is currently being investigated as a target for CAR T-cell therapy of human cancers[15–17]. Therefore, CD276 is an additional target for CAR T therapy against RMS.

Second generation CAR designs allow for extensive customization with selection of single-chain variable fragment (scFv) binder, hinge transmembrane (HTM), and intracellular co-stimulatory domains (CSD) which collectively determine the activation threshold and signaling properties of a CAR[21–24]. We recently reported an FGFR4-targeting CAR, based on a second-generation CAR design with a CD8 HTM and a 4-1BB CSD, that will be tested in a phase I clinical trial[7]. Despite its potent antitumor activities, this FGFR4 CAR's therapeutic effect is unclear in stress conditions such as with a low infused CAR T-cells to tumor burden ratio. Here, we first examined the efficacy of our previously reported FGFR4 CAR T-cells under stress conditions and attempted to improve their potency by replacing HTM and CSD domains with those of CD28. Replacing the 4-1BB with a CD28 CSD resulted in a more rapid tumor eradication but at a price of decreased persistence and increased exhaustion for CAR T-cells as suggested by other studies[25–27].

Despite improvement of single CAR's potency, multi-targeting CARs potentially prevent tumor escape resulting from heterogeneous expression of antigens or loss of target expression under selection pressure[17,28], a phenomenon observed after CAR therapies targeting single antigens[29,30]. However, the optimal strategy for developing dual-targeting CARs has not yet been established, especially for solid tumors. Bivalent constructs are more potent and relatively easy to use compared to co-transduction or co-infusion[17,31,32]. We and other groups have demonstrated that a bicistronic "OR" CAR construct encoding two single targeting CAR cassettes performed better than a bivalent CAR construct containing two tandem scFvs, partly due to the maintenance of structural integrity of assembled antigen-binding moieties[17,32]. Furthermore, the optimal combination of T-cell CSDs from two different CAR cassettes in a bicistronic CAR construct needs to be tested despite a common perception in the field that dual CD28 and 4-1BB co-stimulation may provide an ideal T-cell activation signaling[28,31,32].

Here, after optimizing the antitumor activity of an FGFR4 CAR using different HTM and CSDs, we test bicistronic CARs (BiCisCAR) targeting both FGFR4 and CD276 and compare their therapeutic efficacy in several RMS in vivo models. We find that a BiCisCAR using two distinct CSDs demonstrated synergistic cytokine production and cytotoxicity through a robust activation of downstream TCR signaling pathways. Persistence and limited exhaustion of T-cells expressing this BiCisCAR suggests that our CAR engineering strategy not only addresses heterogeneous expression of target antigens but also provides a robust CAR for treating RMS.

## Results

### Altering HTM or CSD significantly improves FGFR4 CAR activity against RMS orthotopic xenografts

We previously reported that 10 million FGFR4 CAR T-cells with a CD8α HTM and a 4-1BB CSD (FGFR4.8HTM.BBz, Fig. 1A, left) could effectively eradicate an aggressive RH30 FP-RMS orthotopic intramuscular xenografts in NSG mice[7]. However, under stress conditions with only 2.5 million CAR T-cells, tumors cannot be eliminated in this model (Supplementary Fig. 1A–E). Since modifications of CAR HTM and CSD may enhance CAR activities[22,23], we substituted these domains of the original FGFR4.8HTM.BBz CAR with a CD28 HTM (FGFR4.28HTM.BBz) or together with a CD28 CSD domain (FGFR4.28HTM.28z) (Fig. 1A). We tested their efficacy on two RMS xenograft models, RH30 (FP-RMS),

and a more aggressive RMS559 (FN-RMS with a FGFR4 V550L activation mutation[33]). There were modest differences in CAR surface expression levels and CAR+ percentages for the different domains, but no differences CD4/CD8 ratio or memory T-cell differentiation phenotypes of the CAR T-cells (Supplementary Fig. 2A–G). FGFR4.28HTM.BBz showed similar in vitro tumor-killing ability and cytokine production as the original FGFR4.8HTM.BBz CAR T-cells (Fig. 1B, C), but exhibited enhanced in vivo antitumor activity (Fig S1, B to E; Fig. 1F). FGFR4.28HTM.28z demonstrated significantly increased cytotoxicity (Fig. 1B) and cytokine production compared to the other two designs (Fig. 1C; Supplementary Fig. 2H, I). In an RH30 orthotopic mouse model, only 2.5 million of FGFR4.28HTM.BBz or FGFR4.28HTM.28z CAR T-cells could mediate robust responses (Fig. 1D–G) with significant survival benefits (Supplementary Fig. 3A–E), demonstrating their superiority over the original FGFR4-CAR in these stress conditions. In addition, FGFR4.28HTM.28z CAR exhibited better tumor control than FGFR4.28HTM.BBz CAR in a more aggressive orthotopic model of RMS559 cells (Fig. 1H–K; Supplementary Fig. 3F–G), whereas even 10 million of FGFR4.8HTM.BBz CAR T-cells failed to show therapeutic benefit (Supplementary Fig. 4A–D).

We investigated the expansion and propensity for exhaustion of the CAR T-cells expressing the different CSDs in mouse blood. FGFR4.28HTM.BBz displayed better persistence compared to the original FGFR-CAR design using CD8 HTM at day 31 post CAR T-cells infusion in the RH30 orthotopic model (Supplementary Fig. 1F). Furthermore, it also exhibited better persistence compared to FGFR4.28HTM.28z CAR at day 23 post-CAR T-cells infusion in the RH30 model (Fig. 1L; Supplementary Fig. 5A–C) or at day 32 in the RMS559 IM model (Supplementary Fig. 5D–G). Notably, FGFR4.28HTM.28z exhibited higher expression of the T-cell exhaustion markers such as CD39, PD-1, LAG-3, and Tim-3 (Fig. 1M, Supplementary Fig. 5H), in keeping with previous reports that CD28 CSD-based CAR T-cells are susceptible to exhaustion[34–37]. Taking these data together, the CD28 HTM domain imparted a greater tumor-killing activity and better persistence than a CD8 HTM in the second-generation CAR design with a 4-1BB CSD. In addition, although CD28 CSD was linked with a shorter persistence and increased T-cell exhaustion compared to 4-1BB CSD constructs, an FGFR4 CAR using both CD28 HTM and CSD demonstrated significantly enhanced anti-tumor activity, leading to extended survival compared to the 4-1BB-based CARs in a more aggressive RMS559 model.

### Identification and quantification of surface antigens FGFR4 and CD276 in RMS cells

The limited efficacy of improved FGFR4 CAR against RMS559 tumor may be due to the heterogeneous expression of FGFR4 expression on RMS. Therefore, we attempted to identify another tumor-associated antigen to achieve better tumor control by preventing immune escape. CD276 has been reported as a potential downstream target of PAX3-FOXO1 in FP-RMS[20]. Reanalysis of chromatin immunoprecipitation (ChIP) sequencing data[10,38] confirmed CD276 as a target of PAX3-FOXO1 in FP-RMS cell lines (RH4 and RH30), and readily detected both H3K27ac marks and the core regulatory master transcription factor, MYOD1, binding at the enhancers in the *CD276* locus for both FP-RMS and FN-RMS cell lines including RH4, RH30, CTR and RD (Fig. 2A), suggesting active expression of CD276 in RMS cells. Therefore, like FGFR4, CD276 is a biologically relevant cell surface immune target for RMS.

Using RNA-Seq data for human RMS tumors ($n = 98$) and cell lines ($n = 33$), we confirmed that *FGFR4* and *CD276* were highly expressed in RMS compared to 147 normal tissues with significant heterogeneity among tumor samples (Fig. 2B). To determine whether this expression pattern is consistent with their protein level, we quantified the cell surface expression of FGFR4 and CD276 on RMS cell lines and patient-derived xenografts (PDX) using flow cytometry (Fig. 2C–F). On

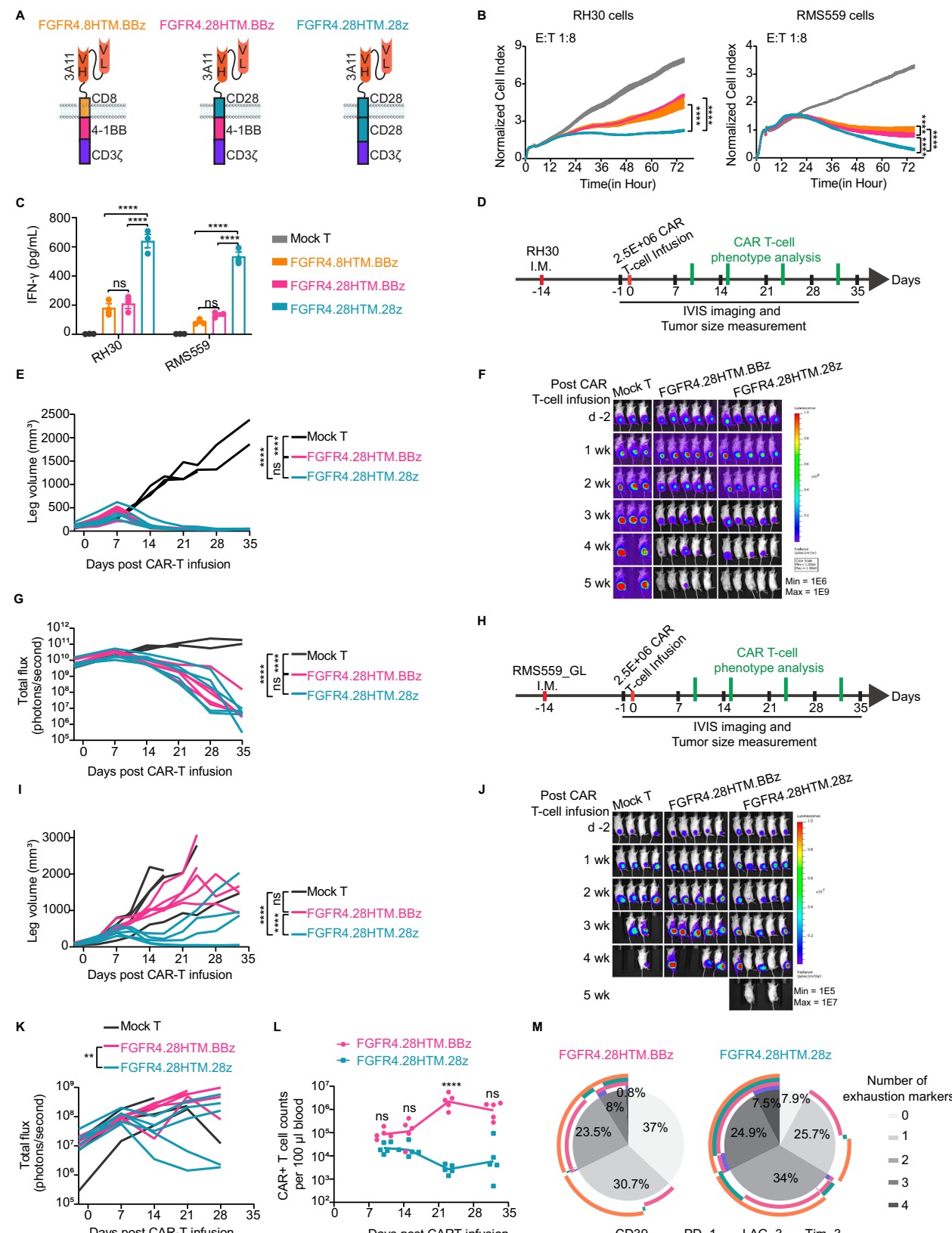

average, RMS cell lines and cell-line-derived xenografts (CDX) expressed 6150 (range 211–15,115) FGFR4 molecules/cell and 41,577 (range 2893–67,786) CD276 molecules/cell respectively, while expression of both antigen on RH30, JR or RMS559 CDXs are remarkably lower than that on their corresponding cell lines (Fig. 2D). Additionally, we found that there were on average 1614 (range 132–6266),

and 3957 (range 1908–6169) molecules/cell for FGFR4 and CD276 respectively on 7 RMS PDXs (Fig. 2E, F). Notably, there was considerable intratumor and interpatient variability in the expression of both antigens in all RMS samples (Fig. 2D, F). Thus, these data suggest that targeting both antigens simultaneously may be more effective to prevent immune escape in CAR T-cell therapies for RMS.

**Fig. 1 | Altering HTM and CSDs significantly improves FGFR4 CAR. A** FGFR4 CARs with different hinge and transmembrane (HTM) or co-stimulatory domains (CSD). **B** A representative cytotoxicity data from three independent experiments using T-cells from 3 donors at an E:T ratio of 1:8. Two-way repeated measures (RM) ANOVA was performed. ****$p < 0.0001$. **C** IFN-γ released by FGFR4 CAR T-cells after a 72-h coculture with RMS cells. Means and SEM of three independent experiments are plotted. ****$p < 0.0001$, ns not significant, by two-way ANOVA with Tukey's multiple-comparison test. **D** Schema of testing CAR T-cells from donor 3 in an RH30 intramuscular (I.M.) xenograft model. Mock or FGFR4 CAR T-cells were infused after 14 days when tumors reached a mean size represented by leg volume of 200 mm³. **E** Leg volume in an RH30 I.M. xenograft model. Each line represents a mouse ($n = 3/5$ per group). Two-way RM ANOVA or mixed-effects analysis is used for $p$ values. ****$p < 0.0001$; ns for not significant. Bioluminescent images (**F**) or total

flux (photons/second, **G**) of RH30 tumor burden. Two-way RM ANOVA or mixed-effects analysis was used. ****$p < 0.0001$; ns not significant. **H** Schema of testing 2.5E6 FGFR4 CAR T-cells from donor 3 in an RMS559 I.M. xenograft mouse model. **I** Leg volume was measured to represent tumor size ($n = 4/5$ per group). ****$p < 0.0001$; ns not significant, by mixed-effects analysis between two groups. Bioluminescent images (**J**) and total flux (photons/second, **K**) of RMS559 tumor burden. **$p = 0.0026$, by mixed-effects analysis. **L** Expansion of FGFR4 CAR T-cells represented as counts in 100 µl blood from mice bearing RH30 I.M. tumors treated with 2.5E6 CAR T-cells at day 10, 15, 23, and 32 ($n = 5$). Each dot represents a mouse. Two-way ANOVA with Sidak's multiple-comparison test was performed at each time point. ****$p < 0.0001$. **M** Percentage of CAR T-cells expressing exhaustion markers CD39, PD-1, LAG-3, and TIM-3 at day 32 after T-cell infusion into RH30-bearing mice. Source data is provided as a Source Data file.

## Dual targeting FGFR4 and CD276 BiCisCARs eradicate RH30 and RMS559 orthotopic tumors

Given high CD276 expression on RMS cells, we tested the efficacy of a previously published CD276-targeting construct (CD276.8HTM.BBz; Fig. 3A) against RMS in an RH30 I.M. xenograft model. We found that CD276 CAR T-cells significantly shrank tumors, however, they could not eliminate tumors in all treated mice (Fig. 3B–D).

To address the heterogeneous expression of target antigens that may lead to resistance, we generated 3 bicistronic lentiviral constructs encoding 2 CARs to target both FGFR4 and CD276, separated by a self-cleaving linker allowing co-expression of both CARs in the same T-cell[17]. We used 3 FGFR4 single CARs (FGFR4.8HTM.BBz, FGFR4.28HTM.BBz, and FGFR4.28HTM.28z) in combination with the CD276 CAR (CD276.8HTM.BBz) to construct 3 BiCisCARs respectively (Fig. 3E) and then tested their properties. T-cells expressing three BiCisCARs had similar comparable CAR-transduction efficiencies, FGFR4 and CD276 binding capacity, CD4/CD8 ratios, or memory T-cell differentiation phenotypes (Supplementary Fig. 6A–F). Moreover, they exhibited a similar high tumor-killing ability against RH30 in vitro at an effector: tumor (E: T) ratio of 1:10 (Supplementary Fig. 7A). However, FGFR4.8HTM.BBz-CD276.8HTM.BBz BiCisCAR, which shares the same CD8 HTM and 4-1BB CSD for both FGFR4 and CD276 CAR cassettes, only delayed tumor progression compared to mock T-cells in an RH30 orthotopic xenograft model (Supplementary Fig. 7B–F). We observed a reduced ability of this CAR transduced T-cells for persistence compared to those expressing the other two BiCisCARs (Supplementary Fig. 7G). In contrast, both FGFR4.28HTM.BBz-CD276.8HTM.BBz and FGFR4.28HTM.28z-CD276.8HTM.BBz BiCisCARs completely eradicated tumors in all mice, and the latter showed a more rapid tumor elimination (Supplementary Fig. 7B–F).

To further evaluate these two more effective BiCisCARs, we used aggressive RMS559 cell in vitro and in vivo models where FGFR4 single CAR T-cells showed limited antitumor activity. The FGFR4.28HTM.28z-CD276.8HTM.BBz BiCisCAR T-cells demonstrated superior tumor-killing activity than those of FGFR4.28HTM.BBz-CD276.8HTM.BBz in vitro (Fig. 3F). When tested in vivo, T-cells expressing FGFR4.28HTM.28z-CD276.8HTM.BBz BiCisCAR exhibited faster tumor eradication, increased persistence, and reduced expression of exhaustion markers such as CD39, PD-1, and LAG-3 (Fig. 3G–L).

## FGFR4.28HTM.28z-CD276.8HTM.BBz BiCisCAR showed superior expansion, tumor infiltration, and limited exhaustion under stress conditions

We next tested these CARs including two FGFR4-targeting CARs (FGFR4.28HTM.BBz and FGFR4.28HTM.28z), one CD276-targeting CAR (CD276.8HTM.BBz), and two BiCisCARs (FGFR4.28HTM.BBz-CD276.8HTM.BBz and FGFR4.28HTM.28z-CD276.8HTM.BBz) in a high-stress model where only 1 million CAR T cells were used to treat aggressive RMS559 orthotopic xenografts (Fig. 4A). Two FGFR4-targeting CARs and FGFR4.28HTM.BBz-CD276.8HTM.BBz BiCisCAR showed minimal antitumor activity, and CD276-targeting CAR cleared

tumors in 2 out of 5 mice. By contrast, FGFR4.28HTM.28z-CD276.8HTM.BBz BiCisCAR demonstrated the highest potency, controlling tumors in 4 out of 5 mice, accompanied by a more substantial cell expansion (Fig. 4B–F).

Because of very high-level expression of CD276 on both RH30 and RMS559 cells and dose-dependence on their antigen target density on tumors for CAR T-cell activity[22], we next examine the performance of these CARs using another aggressive FP-RMS RMS cell JR expressing lower CD276 ($8066 \pm 123$ molecules/cell in cell line; $2893 \pm 234$ molecules/cell in xenograft) (Fig. 2D). Luciferase-labeled JR cells were intramuscularly injected in NSG mice and then treated with 2.5 million CAR T-cells 13 days later (Fig. 4G). Similar to the results in the aforementioned RMS559 model under stressed conditions, the two single targeting FGFR4 CAR T-cells failed to control tumor growth (Fig. 4H). FGFR4.28HTM.BBz-CD276.8HTM.BBz BiCisCAR T-cells only had modest antitumor activity, worse than the single CD276 targeting CAR (Fig. 4H, I). Nevertheless, both the CD276.8HTM.BBz single CAR and FGFR4.28HTM.28z-CD276.8HTM.BBz BiCisCAR T-cells controlled tumor and significantly increased survival, with the BiCisCAR demonstrating a swifter tumor clearance (Fig. 4H, I; Supplementary Fig. 8A).

T-cell expansion, tumor infiltration, and exhaustion are pivotal factors that influence the therapeutic efficacy of CAR T-cells[24,39–41]. To gain a deeper insight into these characteristics of our CARs, we analyzed tumor-infiltrating CAR T-cells and circulating CAR T-cells in the blood on day 16 or 21 using flow cytometry before tumor clearance. Remarkably, the FGFR4.28HTM.28z-CD276.8HTM.BBz BiCisCAR T-cells exhibited the highest CAR T-cell count in the bloodstream and within tumors when compared to the other four CARs, including CD276.8HTM.BBz (Fig. 4J, K). Furthermore, the proportion of T-cells transduced with FGFR4.28HTM.28z-CD276.8HTM.BBz BiCisCAR that expressed exhaustion markers was significantly lower within tumor (CD39 and Tim-3) and in the blood (CD39 and LAG-3) compared to those observed with other CARs (Fig. 4L–O; Supplementary Fig. 8B–E). These findings are consistent with the enhanced and rapid antitumor activity of this BiCisCAR. Consequently, FGFR4.28HTM.28z-CD276.8HTM.BBz BiCisCAR surpassed all other single-targeting CARs and dual-targeting BiCisCARs, showing the most robust expansion and the least exhaustion.

## Multimodal single-cell profiling reveals a distinct phenotype of FGFR4.28HTM.28z-CD276.8HTM.BBz BiCisCAR T-cells, confirming its functional superiority over other CAR T-cells

To gain insight into the molecular underpinnings of the superior antitumor effect exhibited by FGFR4.28HTM.28z-CD276.8HTM.BBz BiCisCAR T-cells, we performed Cellular Indexing of Transcriptomes and Epitopes by sequencing (CITE-Seq) using 14 protein markers (Supplementary Table 1) on tumor-infiltrating T-cells (TIL) isolated from JR tumor-bearing mice 11 days after CAR T-cells treatment. This time point coincided with the onset of maximal infiltration of TILs, just before observable tumor shrinkage (Fig. 5A). After filtering and batch effect correction, we obtained high-quality data for 38,108 tumor-infiltrating T-cells from 10 mice. We identified 13 distinct cell

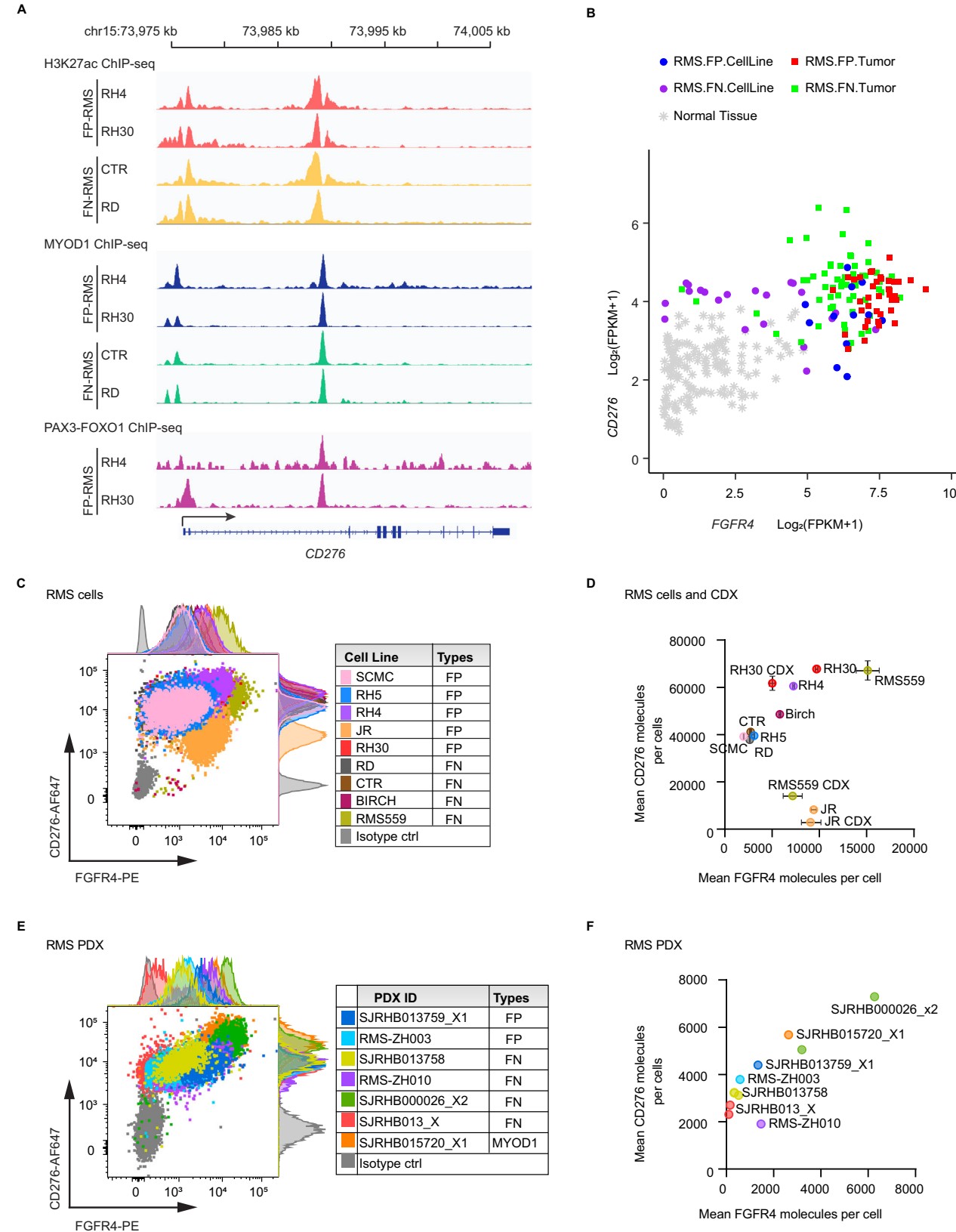

populations using weighted nearest-neighbor (WNN) analysis (see Methods) of integrated protein and transcriptome data (Supplementary Fig. 9A, B). These cell populations include CD8+ T-cells (clusters 1, 2, 3, 5, 7, 10, and 14), CD4+ T-cells (clusters 0 and 9), CD4+ or CD8+ mixture T-cells (cluster 6), γδ T-cells (cluster 11), and natural killer (NK)-like T-cells (cluster 18) (Supplementary Fig. 9B, C; Supplementary

Fig. 10). Expression signatures of these clusters allowed us to annotate each T-cell subtype (Supplementary Table 2, Supplementary Data 1, Supplementary Fig. 9C and Supplementary Fig. 10).

TILs exhibited notable differences of subtypes among mice treated with various CAR T-cells (Fig. 5B, C). For example, CD276.8HTM.BBz CAR T-cells demonstrated the highest fraction of activated T effector

**Fig. 2 | Direct targeting and establishment of enhancers at the *CD276* locus by PAX3-FOXO1 and MYOD1, and heterogenous expression of FGFR4 and CD276 on RMS cell lines, CDXs or PDXs. A** H3K27ac (top), MYOD1 (middle), and PAX3-FOXO1 (bottom) ChIP-seq at the CD276 locus in FP-RMS cell lines (red, dark blue, and purple, respectively) and FN-RMS cell lines (orange, green). **B** Scatterplot of FGFR4 and CD276 mRNA demonstrated a generally high level of expression for CD276 but more variable expression of FGFR4 in RMS tumors or cells compared to normal tissues. **C–F** Flow cytometry was used to measure FGFR4 or CD276 expression on patient-derived RMS cell lines (**C**) or patient-derived xenografts (PDX, **E**) by staining with anti-human FGFR4 antibody (3A11) and anti-human CD276 antibody (MGA271). Co-staining of FGFR4 and CD276 showed heterogeneity of expression of both targets on RMS cells and PDXs. Cell line or PDX IDs and tumor types are listed in the right tables (**C** and **E**). Quantification of FGFR4 or CD276 molecules on each RMS cell and CDX (**D**) or PDXs (**F**) was performed using a phycoerythrin (PE) fluorescence quantitation kit. Dots with error bars representing mean ± SD in Figure **D** show protein expression measured by flow cytometric analysis from 3 independent experiments. Source data is provided as a Source Data file.

cells (C6_Activated Teff, IFNG+) in both CD4+ and CD8+ compartments (Fig. 5C). Interestingly, the FGFR4.28HTM.28z-CD276.8HTM.BBz BiCisCAR displayed a higher proportion of CD8+ effector memory cells and granzyme-positive effector T-cells (C1 + C3 + C7) within the CD8+ T-cell populations, as depicted in Fig. 5B, C. Concurrently, tumors treated with CD276.8HTM.BBz single CAR and FGFR4.28HTM.28z-CD276.8HTM.BBz BiCisCAR T cells exhibited lower frequencies of activated CD4+ T-cells (Fig. 5B, C). These findings are consistent with the established differentiation program of T-cell responses following activation (activated T→IFN-γ producing activated Teff→ cytotoxic T-cell)[42–45]. Furthermore, to identify genes that were preferentially expressed in tumor-infiltrating T-cells expressing FGFR4.28HTM.28z-CD276.8HTM.BBz BiCisCAR, we performed a differential expression analysis (Fig. 5D). This analysis revealed the overexpression of genes such as perforin and multiple granzymes, which are associated with antitumor activity and are considered hallmarks of cytotoxic CD8+ T-cells (Fig. 5E). Therefore, these data demonstrate that FGFR4.28HTM.28z-CD276.8HTM.BBz BiCisCAR T-cells rapidly expanded to generate effector and memory subsets in vivo, explaining their heightened potency against the aggressive RMS559 and JR solid RMS tumors.

### BiCisCAR T-cells overcome heterogeneous expression of FGFR4 and CD276 in vivo

To assess the BiCisCAR's capacity to address the heterogeneous expression of FGFR4 and CD276 in RMS, we created CRISPR-KO clones for FGFR4 or CD276 in RH30 cells (Fig. 6A). As anticipated, single CAR T-cells could not eliminate RH30 cells lacking the target antigen. In contrast, FGFR4.28HTM.28z-CD276.8HTM.BBz BiCisCAR T-cells demonstrated consistent cytotoxicity regardless of the presence or absence of either FGFR4 or CD276 (Fig. 6B–D).

To model heterogeneous expression of tumor antigens in an in vivo model, we established bilateral orthotopic xenografts with RH30 FGFR4-KO (right leg) and CD276-KO (left leg) (Fig. 6E). Consistent with our in vitro data, mice treated with single CAR T-cells exhibited selective tumor shrinkage or eradication only in the presence of the corresponding target antigen. In contrast, dual-targeting CAR T-cells effectively eliminated tumors on both sides (Fig. 6F–J). Notably, FGFR4 single-targeting CAR T-cells only eliminated 3 out of 5 FGFR4+ tumors, whereas BiCisCAR eradicated all FGFR4+ tumors, indicating a heightened antitumor activity of BiCisCAR against FGFR4+ RH30 tumors. This observation is consistent with the characteristics of these CARs in terms of persistence and exhaustion (Fig. 1L, M; Fig. 4J–O). Therefore, this experiment not only affirmed the specificity of single CARs but also underscored the efficacy of BiCisCAR in eradicating tumors with heterogeneous antigen expression.

### Presence of both CD28 and 4-1BB CSDs in dual-targeting CAR T-cells leads to heightened CAR signal strength, sustained T-cell activation signaling

Our data, in conjunction with findings from previous studies[25,26,36,37,46], indicate that CAR T-cells with a CD28 CSD exhibit rapid expansion and increased anti-tumor activity but are linked to shorter persistence and a greater tendency for T-cell exhaustion compared to those with a 4-1BB CSD. T-cells expressing dual targeting FGFR4.28HTM.28z-CD276.8HTM.BBz BiCisCAR demonstrated the highest level of tumor killing, T-cell expansion, persistence, and lowest degree of exhaustion. This suggests an additive or potentially synergistic effect resulting from the presence of both CD28 and 4-1BB CSDs in the same T-cell. To quantify the activity of the BiCisCAR, we first compared its cytotoxicity to single CARs in vitro. T-cells expressing the FGFR4.28HTM.28z BiCisCAR exhibited superior cytotoxic activity than the single-targeting CARs against RH30, RMS559, and JR cells which express high levels of FGFR4 but varying levels of CD276 (Supplementary Fig. 11). We then measured the cytokine production in activated BiCisCAR T-cells following stimulation with serially diluted plate-coated proteins containing FGFR4, CD276, or both antigens. The production of all three cytokines, IFN-γ, IL-2, and TNF-α, displayed synergistic behavior when both CAR binders engaged their respective antigens for the FGFR4.28HTM.28z-CD276.8HTM.BBz BiCisCAR T-cells at the antigen concentration above 0.00625 μM (Fig. 7A–C). In contrast, FGFR4.28HTM.BBz-CD276.8HTM.BBz BiCisCAR T-cells exhibited lower cytokine levels following dual activation compared to CD276 activation alone at antigen concentrations <0.01 μM, with only weak synergistic effects on IFN-γ and IL-2 production and an additive effect for TNF-α production even at higher antigen concentration (>0.025 μM) (Supplementary Fig. 12A–C). Hence, these data suggest BiCisCAR T-cells possessing both CD28 and 4-1BB CSDs exhibit a strong synergistic effect for cytokine production, whereas the BiCisCAR using the same 4-1BB CSD domain primarily showed additive or modest synergistic effects at higher antigen densities.

The activation of TCR triggers signaling cascades that ultimately influence cell fate by regulating cytokine production, cell survival, proliferation, and differentiation[47,48]. We hypothesized that dual-targeting CARs with different CSDs activate distinct signaling pathways, resulting in optimal signaling strength for robust T-cell activation. To test this hypothesis, we examined the downstream phosphorylation of the two BiCisCARs in T-cells stimulated by soluble FGFR4-Fc, CD276-Fc, or both antigens at various time points. Western blotting revealed higher phosphorylation levels of proximal signaling proteins (pCD3ζ-CAR, pZAP70, and pPLCγ1) and distal signaling proteins (pAKT and pERK1/2) in FGFR4.28HTM.28z-CD276.8HTM.BBz BiCisCAR T-cells following FGFR4 stimulation compared to CD276 stimulation for 5 or 15 minutes (Fig. 7D; Supplementary Fig. 13). In addition, sustained phosphorylation of most of these signaling proteins was observed at 30 min post-stimulation in the presence of both antigens compared to single antigen stimulation (Fig. 7D; Supplementary Fig. 13). Notably, we also observed a higher level of p65 phosphorylation, which is associated with 4-1BB signaling[49], in CD276 CAR T-cells compared to CD28-based FGFR4 CAR T-cells at 5 min after antigen stimulation (Fig. 7D; Supplementary Fig. 13). Dual stimulation in FGFR4.28HTM.28z-CD276.8HTM.BBz BiCisCAR T-cells further prolonged p65 phosphorylation to at least 30 min post-stimulation (Fig. 7D). In contrast, the two 4-1BB-based CAR cassettes in FGFR4.28HTM.BBz-CD276.8HTM.BBz BiCisCAR T-cells showed lower levels of PLCγ1, AKT, and ERK1/2 phosphorylation stimulated by FGFR4 or CD276 protein for 5 min (Supplementary Fig. 12D). Additionally, the FGFR4.28HTM.28z-CD276.8HTM.BBz BiCisCAR exhibited higher activation of PLCγ1, AKT, and ERK1/2 phosphorylation compared to FGFR4.28HTM.BBz-CD276.8HTM.BBz BiCisCAR T-cells upon dual stimulation with both FGFR4 and CD276 (Supplementary Fig. 12D). Thus,

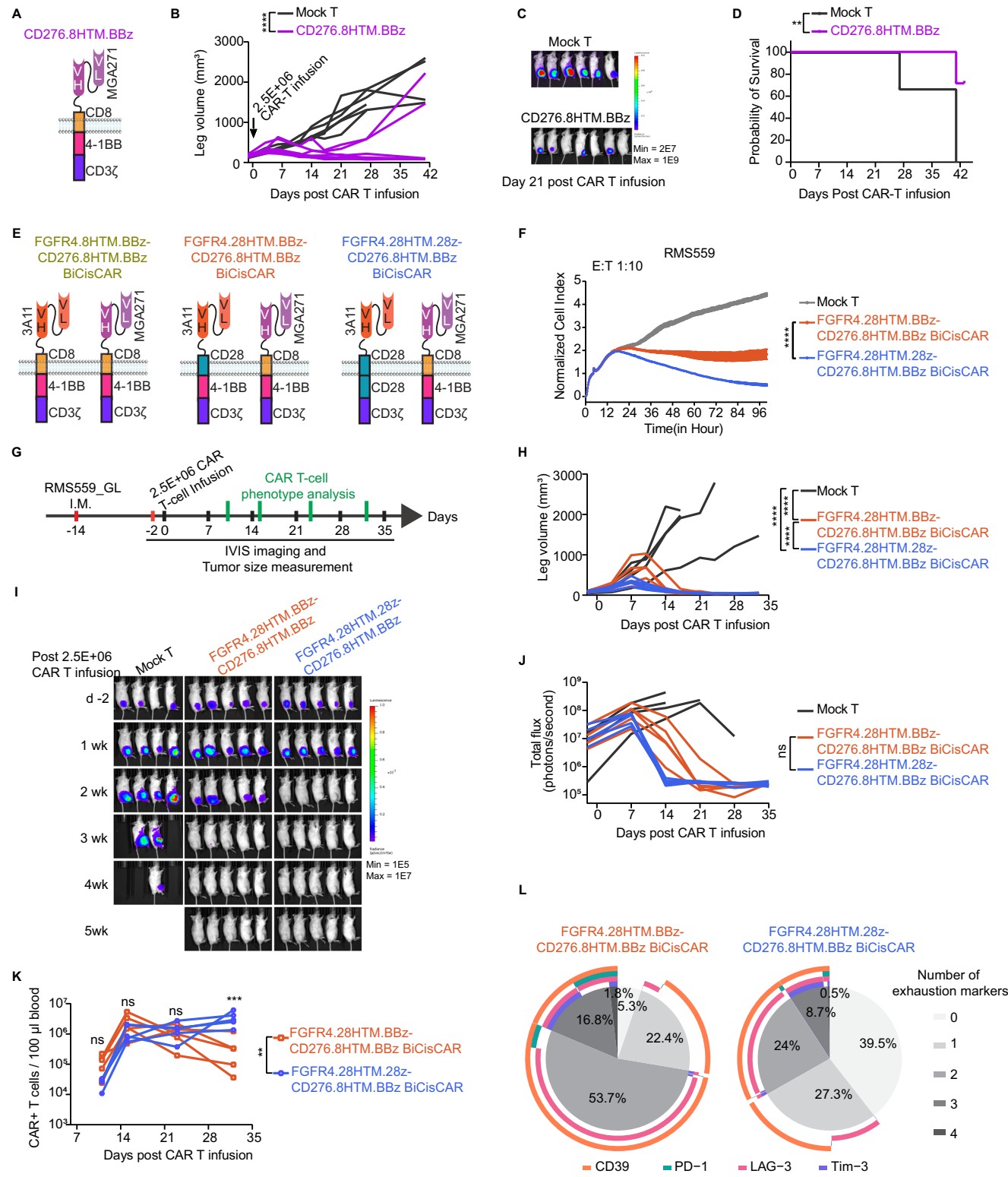

the presence of two different CSDs in the BiCisCAR T-cells likely initiates complementary signaling pathways downstream of TCR, including AKT, ERK1/2, and p65 (Fig. 7E). Activation of these pathways enhances T-cell activation response and may consequently augment CAR T-cell antitumor activity.

## Discussion

The objective of this study was to improve the efficacy of CAR T-cells against solid tumors and to develop principles for dual targeting CAR using RMS as a model. We found that replacing the CD8 hinge and transmembrane domain (HTM) and the 4-1BB co-stimulatory domain (CSD) of our recently reported clinical grade FGFR4 targeting CAR with CD28 HTM and CSD resulted in increased cytotoxicity in vitro and improved anti-tumor efficacy in vivo. However, these substitutions were associated with shorter persistence and increased exhaustion, in keeping with previous studies[22,23].

Simultaneous targeting of two tumor-associated cell surface antigens by a bicistronic CAR represents a promising therapeutic

**Fig. 3 | Dual targeting CAR T-cells using distinct CSDs exhibit faster tumor killing with persistence and limited exhaustion. A** Schematic a CD276 targeting CAR, CD276.8HTM.BBz, containing a CD8HTM and 4-1BB CSD. **B** Leg (tumor) volumes of mice bearing RH30 I.M tumors after 2.5E + 6 CD276.8HTM.BBz CAR T-cell infusion at day 7 post-implantation. Each line represents tumor volume in a mouse (*n* = 6 in Mock-T group, *n* = 7 in CD276.8HTM.BBz group). ****p < 0.0001, determined by two-way RM ANOVA. **C** Representative bioluminescence images of RH30 tumor on day 21 post-CAR T-cells infusion. **D** Kaplan–Meier survival analysis of RH30 mouse model. **p = 0.0069, by a log-rank test. **E** Schematic FGFR4 and CD276 dual targeting BiCisCARs. **F** Cytotoxicity of two BiCisCAR T-cells against RMS559 at an effector (E): target (T) ratio of 1:10. Representative of 3 independent experiments with T cells from 3 donors. Two-way RM ANOVA was performed. ****p < 0.0001. **G** Schema of testing BiCisCAR T-cells in an RMS559 I.M. xenograft

mouse model. **H** Leg volumes after CAR T-cell infusion. Each line represents tumor volume from a mouse (*n* = 4 in Mock T group, or *n* = 5 in CAR T-cell groups). ****p < 0.0001, as determined by two-way RM ANOVA. Bioluminescence images (**I**) and total flux (**J**) of RMS559 tumor. ns, not significant, determined by two-way RM ANOVA analysis. Mock T-cells cohort is also used in Fig. 1J, as both experiments were performed at the same time. **K** Expansion of BiCisCAR T-cells was analyzed as counts per 100 μl blood after 2.5E6 CAR T-cells infusion by flow cytometry at day 11, 15, 23, and 32 (*n* = 5). **p = 0.0023 as determined by two-way RM ANOVA for comparing two series of BiCisCAR and mock T-cells. Two-way ANOVA with Sidak's multiple-comparison test was performed to compare CAR T-cell counts of two groups at each time point. ***p = 0.0004. **L** Fractions of CAR T-cells expressing exhaustion markers CD39, PD-1, LAG-3, and TIM-3 at day 32 after T-cell infusion in the RMS559 I.M. model. Source data is provided as a Source Data file.

strategy to tackle the heterogeneous expression of target antigens and avoid immune escape in solid tumors[17]. Expression levels of FGFR4 in RMS demonstrated heterogeneity at both the mRNA and protein levels. Subsequently, we identified CD276 as a second cell-surface antigen, which was directly targeted by PAX3-FOXO1 in FP-RMS, and regulated by MYOD1, with the establishment of a super-enhancer and associated with high expression in the majority of FN- and FP-RMS. Thus, our data supported the rationale for dual targeting of both FGFR4 and CD276 as an effective approach for RMS treatment.

We evaluated three dual-targeting BiCisCARs, combining different FGFR4-targeting constructs with a CD276-targeting CAR. The BiCisCAR using FGFR4-targeting CAR with a CD8HTM and a 4-1BB CSD did not show any advantage in an RH30 orthotopic xenograft model, partially due to weak T-cell expansion or persistence. Although the other two BiCisCAR variants exhibited comparable tumor clearance in RH30 and RMS559 IM tumors, the FGFR4.28HTM.28z-CD276.8HTM.BBz BiCisCAR, incorporating different CSDs, not only cleared tumors more rapidly but also exhibited robust antitumor activity in a stressful RMS559 orthotopic model with only 1 million CAR T-cells administered. This BiCisCAR also outperformed the other CARs in another aggressive, JR, I.M. model, suggesting the combination of CD28 and 4-1BB CSDs may be a more effective strategy for dual-targeting CAR design, consistent with CD19/CD22 or BCMA/GPRC5D bicistronic CARs[31,32]. Interestingly, third-generation single CARs containing both CD28 and 4-1BB CSDs in tandem were often reported to be inferior compared with their second-generation counterparts in clinical trials[50,51], possibly due to steric hindrance of cis-CSDs interfering with the interactions of co-activators required for optimal T-cell activation and survival. Although recent findings demonstrated that patients with high-risk refractory or relapsed neuroblastoma responded well to the third generation GD2-CART01 cells[52], third-generation CARs targeting single antigens do not address the challenge of heterogeneous target antigen expression. Additionally, Hirabayashi et al. proposed a GD2/CD276 dual-targeting CAR design with optimal CD28 and 4-1BB co-stimulation containing a single CD3ζ domain[28]. This design aims to mitigate the excessive CD3ζ signaling, potentially providing rapid anti-tumor effects in vivo stress conditions. However, it's noteworthy that immune escape may occur in tumors solely expressing CD276, because CD276-targeting CAR cassette in this design cannot be activated in the absence of CD3ζ-chain. Therefore, our current dual-targeting BiCisCAR design strategy not only provides both CD28 and 4-1BB co-stimulation signaling for improved T-cell activation, but also allows for simultaneous targeting of two tumor-associated antigens to prevent potential tumor escape.

In this study, we also investigated the mechanism behind the observed synergistic signaling resulting from both CARs concurrently engaging with their cognate antigens. We demonstrated that activation of the CD28 CSD of FGFR4 CAR induced higher phosphorylation of cascade signaling molecules, including pZAP70-pPLCγ1-pAKT-mTORC1 and pZAP70-pPLCγ1-pERK1/2-AP-1, consistent with rapid antitumor effects[26,53,54]. Concurrently, activation of 4-1BB CSD of

CD276 CAR increased phosphorylation of p65, subsequently activating the NF-κB pathway, critical for sustaining CAR T-cell persistence[46]. Therefore, our BiCisCAR T-cells combine the advantages of both CD28 and 4-1BB signaling when engaging their cognate antigens, resulting in a stronger, sustained, and complementary TCR activation through mTORC1, AP-1, and NF-κB signaling pathways. Additionally, through a multi-modal single-cell analysis of CAR T-cells isolated from tumors in animal models, we observed a high proportion of effector T-cells and effector memory T-cells expressing high levels of cytotoxic genes (GZMK, GZMA, GZMH, GZMB, GNLY, PRF1, and NKG7) in T-cells expressing FGFR4.28HTM.28z-CD276.8HTM.BBz BiCisCAR. This could explain the potency of this BiCisCAR, consistently displaying the fastest tumor shrinkage or clearance throughout our study, including the eradication of RMS559 tumors with just 1 million CAR T-cells.

Despite demonstrating our BiCisCAR's ability to eliminate aggressive orthotopic RMS tumors in vivo, however, we acknowledge two limitations in this study. First, we have not yet addressed the potential of increased systemic toxicity of the dual-targeting CAR T-cells to normal tissues expressing low levels of the targeted antigens. Second, the use of immunocompromised mice bearing human RMS and T-cells limits our ability to fully comprehend the interaction between dual-targeting CAR T-cells and the native RMS micro-environment. Ultimately, a well-designed phase I clinical trial will be necessary to evaluate the safety and efficacy of this CAR in RMS patients.

In summary, we have refined a dual-targeting BiCisCAR T-cell with two different HTMs and CSDs, which exhibit advantageous intrinsic properties, such as heightened signaling, rapid expansion, augmented differentiation of effector and effector memory T-cells, prolonged persistence, and reduced exhaustion. Concurrently, it effectively addresses the challenge posed by heterogenous target antigen expression. These attributes not only facilitate swift tumor eradication and prolonged efficacy, but also decrease the risk of relapse due to antigen escape. The clinical development of this and other similarly designed BiCisCARs has the potential to provide powerful and effective therapies for patients with aggressive solid cancers.

## Methods

All research conducted for this manuscript complies with ethical regulations including approval by the National Institute of Health's Institutional Animal Care and Use Committee, protocol GB-011.

### Cell lines, cell culture, CDX and PDX tumors

Human RMS cell lines RH30, RH5, and RH4 were all provided by Dr. Peter Houghton, Greehey Children's Cancer Research Institute, San Antonio, Texas, USA. JR was a gift from Dr. Corinne M. Linardic at Duke University School of Medicine, Durham, NC, USA. RMS559 was obtained from Dr. Jonathan Fletcher at Brigham and Women's Hospital, Boston, USA. Dr. Lee Helman from Children's Hospital Los Angeles, CA, USA, kindly provided RD, CTR, and BIRCH. SCMC was provided by Dr. Janet Shipley, Institute of Cancer Research, London, England.

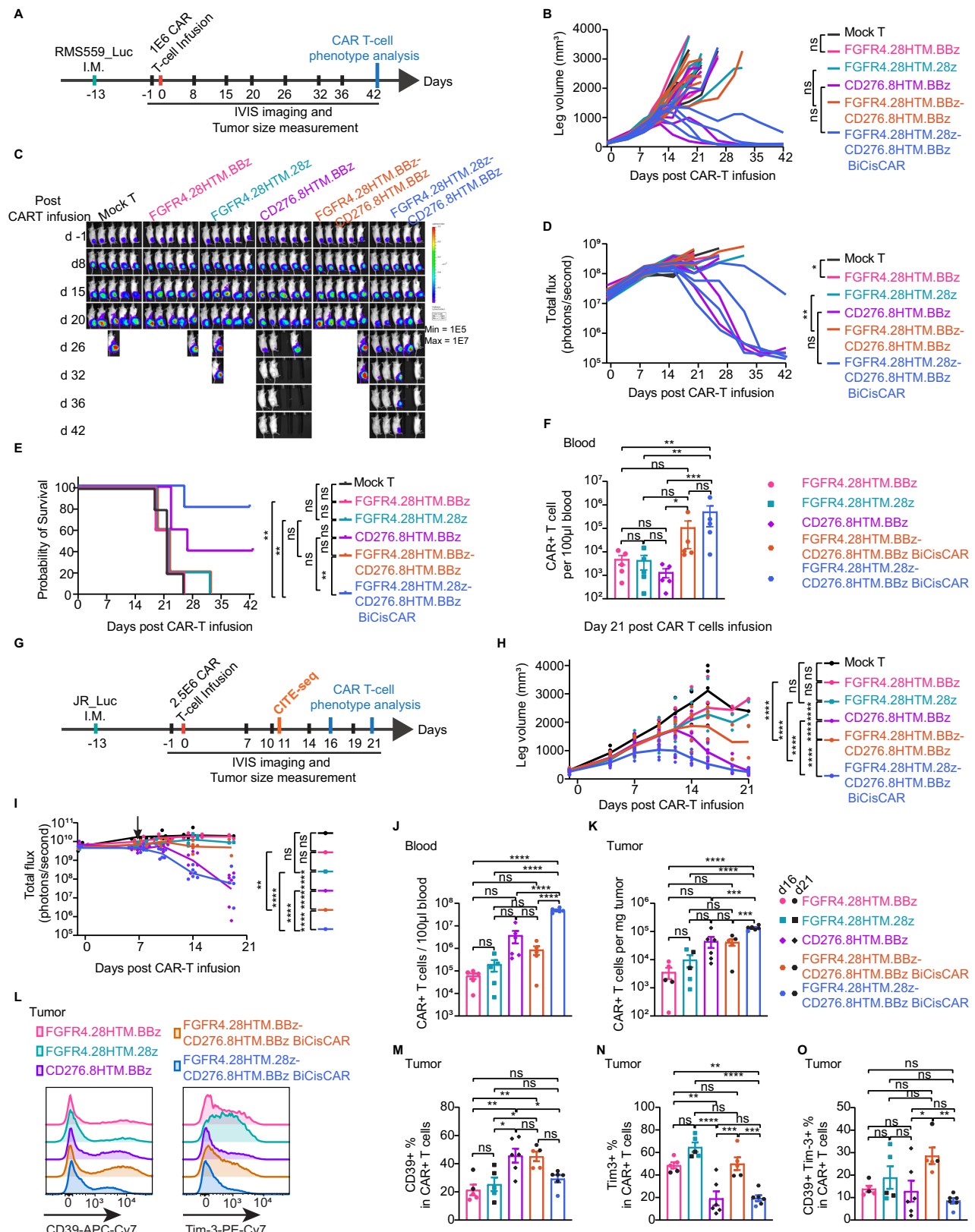

RH30, RMS559, SCMC, RH5, RH4, RD, and CTR cells were cultured in Dulbecco Modified Eagle Medium (DMEM, Quality Biological), supplemented with 10% FBS (Gibco, Life Technologies), 10 mM HEPES, 100 U/mL penicillin, 100 µg/ml streptomycin and 2mM L-glutamine (Gibco, Life technologies). JR and BIRCH cell line were cultured in RPMI1640, supplemented with 10% FBS (Gibco, Life Technologies),

100 U/mL penicillin, 100 µg/ml streptomycin, and 2m M L-glutamine. All cell lines were STR DNA fingerprinted to confirm their identity every 6 months and regularly tested to be mycoplasma negative by a MycoAlert kit (Lonza).

RMS cell lines RH30, RMS559, and JR were stably transduced with a lentiviral vector encoding the GFP-Firefly-Luciferase gene for

**Fig. 4 | Dual targeting BiCisCAR with two different CSDs shows enhanced expansion, tumor-infiltrating and limited exhaustion. A** Schema of testing 1E + 6 CAR T-cells in an RMS559 I.M. mouse model. Tumor size (**B**), bioluminescence images (**C**), and total bioluminescence flux (**D**) were monitored for RMS559_Luc orthotopic xenograft. *$p = 0.05$, **$p = 0.0042$, by mixed-effects analysis. (**E**) Kaplan–Meier survival analysis ($n = 5$ per group). **$p < 0.01$, ns, not significant, by Log-rank test. **F** Mean counts ± SEM of CAR$^+$ T-cells (gating from CD45$^+$CD3$^+$ T-cells) in blood 21 days after CAR T-cell infusion. $n = 5$ per group. One-way ANOVA with Holm-Sidak's multiple comparison tests was performed for Log10(CAR$^+$ T-cell counts). *$p < 0.05$, **$p < 0.01$, ***$p < 0.001$. **G** Schema of testing CAR T-cells in a JR I.M xenograft NSG mouse model. $n = 5$ per treated group, except for CD276.8HTM.BBz CAR and FGFR4.28HTM.28z-CD276.8HTM.BBz BiCisCAR T-cells group ($n = 6$) in figures **H**–**O**. Tumor size (**H**) or bioluminescence kinetics (**I**) for JR_Luc orthotopic

xenografts. **$p = 0.0063$, ****$p < 0.0001$, by mixed-effects analysis. CAR T-cells (CD45$^+$CD3$^+$ CAR$^+$) in blood (**J**) or tumor (**K**) 16 or 21 days after CAR T-cell treatment. Data are shown as individual values (shapes) and mean ± SEM (bar graphs). ***$p = 0.0002$, ****$p < 0.0001$, by ordinary one-way ANOVA with Tukey's multiple comparisons tests. Data from day 16 are shown as colored shapes; while data from day 21 are shown as black shapes in Figures **K**, **M**–**O**. **L** A representative flow cytometry experiment detects CD39 and Tim-3 expression on CAR$^+$ T-cells from JR xenografts at day 21 post-infusion. Percentages of CD39$^+$ (**M**), Tim-3$^+$ (**N**), and CD39$^+$ Tim-3$^+$ (**O**) CAR$^+$ T-cells in tumors from mice 16 or 21 days after CAR T-cells infusion. Values are represented as mean ± SEM. *$p < 0.05$, **$p < 0.01$, ***$p < 0.001$, and ****$p < 0.0001$, by one-way ANOVA with Tukey's multiple comparisons tests. The full lists of $p$ values in **E**, **F**, **M**, **N**, and **O** can be found in the Source Data file. All data is provided in the Source Data file.

monitoring tumor burden by IVIS imaging. We implanted them into NSG mice as cell line-derived xenografts (CDX) for tumor study. gRNA-encoding lentiCRISPRv2 puro vector (Addgene, Cat# 98290) was used to create *FGFR4*-KO (FGFR4-guide: 5′-GATCGTGGAGTGCGCCGCCA A-3′) RH30 cell line, or *CD276*-KO (CD276-guide-1: 5′- CACCGTGGCAC AGCTCAACCTCATC-3′ and CD276-guide-2: 5′- AAACGATGAGGTTG AGCTGTGCCAC-3′) RH30 cell lines by CRISPR/Cas9 gene-editing technology. And then subclones were generated by screening KO clones with comparable levels of CD276 or FGFR4 respectively as parental cell lines.

Patient-derived xenograft (PDX) tumors for antigen density quantification were obtained from St. Jude Children's Research Hospital (Memphis, Tennessee, USA) and then expanded in NSG mice. After dissecting from mice, tumors were processed into homogenous single-cell suspensions via mechanical dissociation using a gentle MACS dissociator (Miltenyi), passage through a 70-micron filter, and followed by 2 washing steps with PBS. PDX samples used in this study are SJRHB013759_X1 (FP-RMS), RMS-ZH003 (FP-RMS), SJRHB013758 (FN-RMS), RMS-ZH010 (FN-RMS), SJRHB000026_X2 (FN-RMS), SJRHB013_X (FN-RMS), and SJRHB015720_X1 (MYOD1 mutant RMS).

### Generation of CAR constructs

All CAR constructs were generated using the pELPS lentiviral transfer vector with an EF-1α promoter. FGFR4-targeting CARs were generated using the FGFR4-specific single-chain variable fragment (3A11 scFv), the CD8α or CD28 hinge and transmembrane domain (HTM), the CD28 or 4-1BB co-stimulatory domain (CSD) and CD3ζ intracellular domain (FGFR4.8HTM.BBz previously described[7], FGFR4.28HTM.BBz and FGFR4.28HTM.28z). CD276-specific CARs were generated using the MGA271 scFv provided by MacroGenics (Rockville, MD), the CD8α HTM domain, the 4-1BB CSDs and CD3ζ intracellular domain (CD276.8HTM.BBz).

The dual targeting CAR constructs were generated by individually combining the CAR cassette encoding FGFR4.8HTM.BBz, FGFR4. 28HTM.BBz or FGFR4.28HTM.28z with CD276.8HTM.BBz CAR linked with a P2A-sequence peptide (FGFR4.8HTM.BBz- CD276.8HTM.BBz, FGFR4.28HTM.BBz-CD276.8HTM.BBz, and FGFR4.28HTM.28z- CD276. 8HTM.BBz). These constructs were generated using restriction cloning (Rapid T4 DNA Ligation Kit, Takara) or using the Seamless Cloning and Assembly Kit (GeneArt, Invitrogen) according to manufacturer's instructions. Any homologous sequences were codon-wobbled to avoid recombination for these bicistronic CAR constructs.

### Lentiviral preparation, transduction, and expansion of human T-cells

The aforementioned CAR-encoding lentiviral supernatant was produced by a transient transfection of the Lenti-X-293T lentiviral packaging cell line with corresponding CAR plasmids, using a previously described method[55]. Concentrated lentivirus for transduction of human T-cells was prepared as previously described[17]. Briefly, buffy coats from 6 healthy donors were obtained from the NIH blood bank

for peripheral blood mononuclear cells (PBMC) isolation using Histo-paque®-1.077 gm/mL (Sigma, Cat# 10771) according to the manufacturer's instructions. PBMCs were activated with CD3 and CD28 microbeads at a ratio of 1:1 (Dynabeads Human T-Expander CD3/CD28, Thermo Fisher Scientific, Cat# 11141D) in AIM-V media (Invitrogen) containing 40IU/mL recombinant IL-2 (rIL-2, Clinigen Inc.) and 5% heat-inactivated FBS for 48 h. Then activated PBMCs were transduced with CAR-expressing lentivirus at a multiplicity of infection (MOI) of 14 twice, followed by CD3/CD28 beads removal. T cells were expanded in fresh AIM-V media with 5% heat-inactivated FBS and 200IU/mL rIL-2. Culture medium was changed every 2 - 3 d until harvest on day 9 or 10. Mock T-cells, un-transduced T-cells (UTD), were treated the same as CAR-transduced T-cells except during the transduction procedure.

### ChIP-Seq data analysis

Previously published ChIP-seq data for histone mark H3K27ac, transcript factor MYOD1 and fusion gene PAX3-FOXO1 in RMS tumors and cell lines (RH4 and RH30 for FP-RMS; CTR and RD for FN-RMS) were used in the analysis[10,38]. All reads were mapped to human genome build hg19 using BWA, and indexed BAMs were converted to compressed TDF format at 25 bp bin resolution after reads extension to the median fragment length (~200 bp extended past each mapped single end of 75 bp reads). Files were visualized in an IGV viewer (https://software. broadinstitute.org/software/igv/download).

### Quantitation of FGFR4 or CD276 molecules on RMS cells or PDXs

Staining for FGFR4 or CD276 expression on RMS cell lines was performed using a mouse anti-human FGFR4 antibody (clone 3A11, in-housed, 1 μg/ml) or a recombinant humanized anti-human CD276 antibody (clone MGA271, Creative Biolabs, Cat# TAB-117CL, 1 μg/ml), followed by incubation with PE-conjugated Goat anti-mouse immu-noglobulin G (IgG) antibody (Biolegend, Cat# 405307, 1:300 dilution) or R-Phycoerythrin AffiniPure F(ab')$_2$ Fragment Goat Anti-Human IgG, Fcγ Fragment specific (Jackson ImmunoResearch Laboratories, Cat# 109-116-170, 1:300 dilution). FGFR4 or CD276 surface molecules per cell were calculated after subtracting background signal emanat-ing from a respective isotype control antibody (Clone MG1-45 for mouse IgG1 isotype control antibody, Clone QA16A12 for human IgG1 antibody, Biolegend, 1 μg/ml) by the Quantibrite PE Quantitation Kit (BD Biosciences, Cat# 340495) according to the manufacturer's protocol.

For quantification of cell surface FGFR4 and CD276 on RMS CDXs and PDXs, xenografted tumors were implanted into the flank of 5-8 weeks old female NSG mice after thawing and washing. When the tumor engraftment grew up to 100-2000 mm$^3$ measured by a caliper, xenograft tumor tissue was mechanically dissociated by the flat end of a plunger after harvesting and passed through 100 μm cell strainer followed by 2 wash steps with PBS to obtain a homogenous single cell suspension. Single tumor cells were processed immediately for staining as forementioned for cell lines. Dead/live dye was used for identi-fying viable tumor cells.

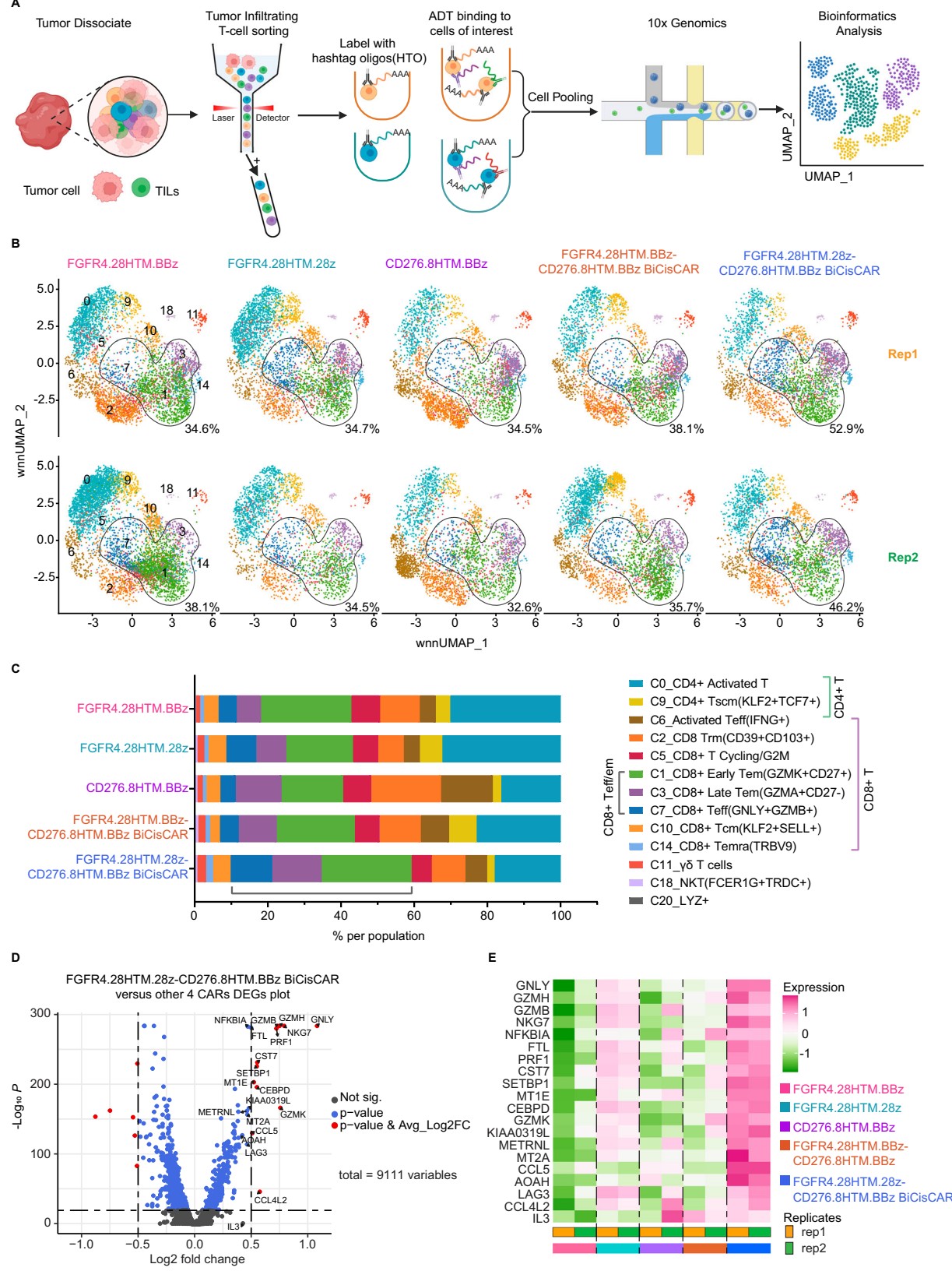

## Cytotoxicity assay of CAR T-cells

xCELLigence® real-time cell analysis (RTCA) was used to test the killing ability of CAR T-cells against tumor cells as previously described[17]. Briefly, 1E + 4 of human RMS cell lines (RH30, RMS559, JR, FGFR4KO or CD276KO RH30 cells) were separately seeded on an E-Plate 16 (ACEABiosciences). After 4 h when cell settling down, CAR T-cells were

added into the corresponding wells at different E:T ratios as indicated in figures. Then tumor cell number was continuously monitored for additional 20, 44, or 68 h and CAR T-cell-mediated cell death was indicated by a decrease in cell number. Data were analyzed using RTCA Software 2.0 (Acea Biosciences) with normalization to that before CAR T-cells addition (about 4 h after tumor cells addition). At the endpoint,

**Fig. 5 | Multimodal single-cell profiling reveals that the FGFR4.28HTM.28z-CD276.8HTM.BBz BiCisCAR exhibits the highest cytotoxicity activity.**
**A** Workflow of CITE-Seq for simultaneous protein and transcript analysis of tumor-infiltrating CAR T-cells at day 11 post-infusion using a JR IM orthotopic model (created with BioRender.com). ADT, antibody-derived tag; HTO, hashtag oligonucleotide. **B** WNN UMAP visualization of tumor-infiltrating T-cells from mice treated using five CAR T-cells with biological replicates. Each dot represents a single cell and cell clusters are labeled by numbers. The black perimeter lines encircle the cell clusters C1, C3, and C7, and the percentages of cells in these clusters are shown. **C** The percentages (means of 2 biological replicates) of each cell subpopulation from five CAR T-cells treated mice. **D** Volcano plot of differentially expressed genes

(DEGs) between the 28HTM.28z-8HTM.BBz BiCisCAR and 4 other CAR T-cells infiltrating in JR I.M xenografts. Genes with an -$\log_{10} P$ (adjusted $P$, two-sided nonparametric Wilcoxon rank-sum test) <20 and |$\log_2$ fold change| > 0.5 are shown in red ($n = 18$). The top 20 highly expressed genes ranked by average $\log_2$ fold change are labeled and associated with T-cell cytotoxicity. **E** Heatmap of these top 20 genes expressed in CAR T-cells isolated from JR I.M xenograft tumors. The FGFR4.28HTM.28z-CD276.8HTM.BBz BiCisCAR T-cells exhibited the highest expression of T-cell cytotoxicity genes among all CAR T cells. The colored scale bar represents z-score values for gene expression. Source data is provided as a Source Data file.

cells were spun down, and supernatant was collected for cytokines production measurement using V-PLEX Custom Human Biomarkers Proinflammatory Panel 1 (Human IFN-γ, Human IL-2, and Human TNF-α, Meso Scale Discovery) following manufactory's instruction.

### Xenograft mouse models

Animal studies (Protocol #GB-011) were approved by the Institutional Animal Care and Use Committee (IACUC) at the NIH and carried out according to the NIH guidelines. Five to eight weeks old female in-house bred NSG mice (NOD.Cg-PrkdcscidIl2rgtm-1Wjl/SzJ; NCI CCR Animal Resource Program, NCI Biological Testing Branch) were used for animal experiments under IVM Microisolator conditions with a light cycle 6 am to 6 pm. Animals bearing engrafted tumors were randomized into cohorts to ensure a similar mean tumor burden/group based on bioluminescent flux[P/s] values and caliper measurements at study enrollment. The maximum size of tumor allowed through our IACUC is one dimension no greater than 1.7 cm at which point mice must be euthanized. This limit was not exceeded in this study. Animals also were euthanized using $CO_2$ inhalation upon showing symptoms of graft versus host diseases (GvHD), including not feeding, lack of activity, abnormal grooming behavior, hunched back posture, or weight loss >10%.

For orthotopic RMS models, luciferase labeled RMS cells (1E + 6 of RH30, 2E + 6 of RMS559, or 3E + 6 of JR, as indicated in figures s) were resuspended in Matrigel (Corning) after washing and intramuscularly injected into NSG mice as previously described[7]. At 7d or 14d after tumor implantation, 1E + 6, 2.5E + 6, or 10E + 6 CAR-positive T-cells were intravenously injected via tail vein. Tumor engraftment and growth were monitored by leg volume measurement using a caliper twice a week and bioluminescence imaging on an IVIS spectrum instrument (Caliper Life Science, Hopkinton, MA, USA) every week. Tumor volumes were calculated as $Leg\ volume = \pi(length \times width^2)/6$. For bioluminescence imaging, mice were intraperitoneally injected (i.p.) with 3 mg D-luciferin (PerkinElmer) and imaged after 20 min. Bioluminescent signal flux was quantified as photons per second per square centimeter per steradian (photons/s/cm$^2$/sr) with Living Image software version 4.5.4 (PerkinElmer, Waltham, MA, USA). Mice were euthanized when either length or width reached/exceeded 1.7 cm or an animal displayed signs of toxicity including excessive weight loss due to tumor burden or GvHD.

### Immunophenotyping staining of circulating, splenic T-cells and TILs by flow cytometry

Mice were bled via submandibular vein at specific intervals (10d, 15d, 23d, or 32d) for characterization of circulating T-cell frequency and/or phenotype. At time of euthanasia, blood, spleen, or tumors were collected for CAR T-cell analyses. Red blood cells (RBC) were first removed from 100 to 200 μL peripheral blood samples using RBC lysis buffer (BioLegend, Cat# 420301). Spleen or xenograft tumor tissue was mechanically minced or dissociated separately by the flat end of a plunger after harvesting and individually passed through 70 μm or 100 μm cell strainer followed by twice wash steps with FACS buffer (DPBS supplemented with 2% heat inactivated FBS and 2 mM EDTA) to

obtain a homogenous single cell suspension. Then cells were stained with the following antibodies specific to human antigens for 30 min in the dark at 4 °C: CD45-FITC (BioLegend, Clone HI30, Cat# 304006, 1:100 dilution), CD62L-Percp/Cy5.5 (BioLegend, Clone DREG-56, Cat# 304824, 1:100 dilution), Tim-3-PE-Cy7 (BioLegend, Clone F38-3E2, Cat# 345014, 1:100 dilution), CD4-PE-Dazzle594 (BioLegend, Clone A161A1, Cat# 357412, 1:100 dilution), CD8-APC (BioLegend, Clone RPA-T8, Cat# 301049, 1:100 dilution), CD3-AF700 (BioLegend, Clone OKT3, Cat# 317340, 1:100 dilution), CD39-APC-Cy7 (BioLegend, Clone A1, Cat# 328226, 1:100 dilution), LAG3-BV421 (BioLegend, Clone 11C3C65, Cat# 369314, 1:100 dilution), CD45RA-Brilliant Violet 605 (BioLegend, Clone HI100, Cat# 304134, 1:100 dilution), PD-1-BV711 (BioLegend, Clone EH12.2H7, Cat# 329722, 1:100 dilution), or CD8-BUV737 (BD, Clone SK1, Cat# 612755, 1:400 dilution). Cells were washed 3 times with FACS buffer before flow cytometry analysis. Cells were gated for viable (fixable viability dye Ghost Dye™ Violet 510 (TONBO, Cat#13-0870-T100, 1:100 dilution) and singlet cells (SSC-W/SSC-H) before assessment of antigen expression. The absolute CAR T-cell counts in the blood or spleen from tumor-bearing NSG mice were quantified using CountBright Absolute Counting beads (Invitrogen, Cat# C36995) on an LSR Fortessa or FACSymphony A5 (BD Bioscience).

CAR expression was assessed by flow cytometry after incubation with soluble, recombinant, human FGFR4-Fc Chimera Protein (Abcam, Cat# ab83999, 1 μg/ml) for FGFR4-CAR T-cells or Biotinylated Human B7-H3 (4Ig)/B7-H3b Protein (Acro Biosystem, Cat# B73-H82F5, 1 μg/ml) for CD276 CAR T-cells, then followed by incubation with R-Phycoerythrin AffiniPure F(ab')2 Fragment Goat Anti-Human IgG, Fcγ Fragment specific (Jackson ImmunoResearch Laboratories, Cat# 109-116-170, 1:300 dilution) and R-Phycoerythrin-conjugated streptavidin or AF647 conjugated Fab specific for human IgG-Fc (Jackson ImmunoResearch Laboratories, Cat# 109-607-008, 1:300 dilution) respectively. Both CAR binders' expression of BiCisCAR T-cells was assessed using a combination of both detection reagents as indicated for individual figures. Data were collected with an LSR Fortessa or FACSymphony A5 (BD Bioscience) using BD FACS DIVA software v8.0.1 and analyzed by FlowJo v10.7.2 software.

### Single-cell analysis of CAR TILs using cellular indexing of transcriptomes and epitopes by sequencing (CITE-Seq)

TILs were purified from RMS JR xenografts 11 days after T-cell administration using 40% and 70% percoll (Cytiva, Cat# 17089102) gradient separation. Viable human CD45$^+$ lymphocytes were separated by sorting on a BD FACSAria Fusion or a BD FACSAria UV. After washing with CITE-seq antibody staining buffer (DPBS with 0.05% BSA), sorted CD45$^+$ lymphocytes were counted, resuspended at a concentration of 1E + 6/50 μL, and stained with TotalSeq-C human "hashtag" antibodies (hashtag 5 or hashtag 6) for each mouse, allowing identification of different replicates from each group in the analysis. Next, the above TILs were individually stained with a cocktail of a TotalSeq-C human lyophilized panel (BioLegend) of 17 surface proteins at a concentration of 0.5 μg per million cells, including CD3 (Clone UCHT1), CD4 (Clone RPA-T4), CD8 (Clone RPA-T8), CD45RA (Clone HI100), CD45RO (Clone UCHL1), CD27

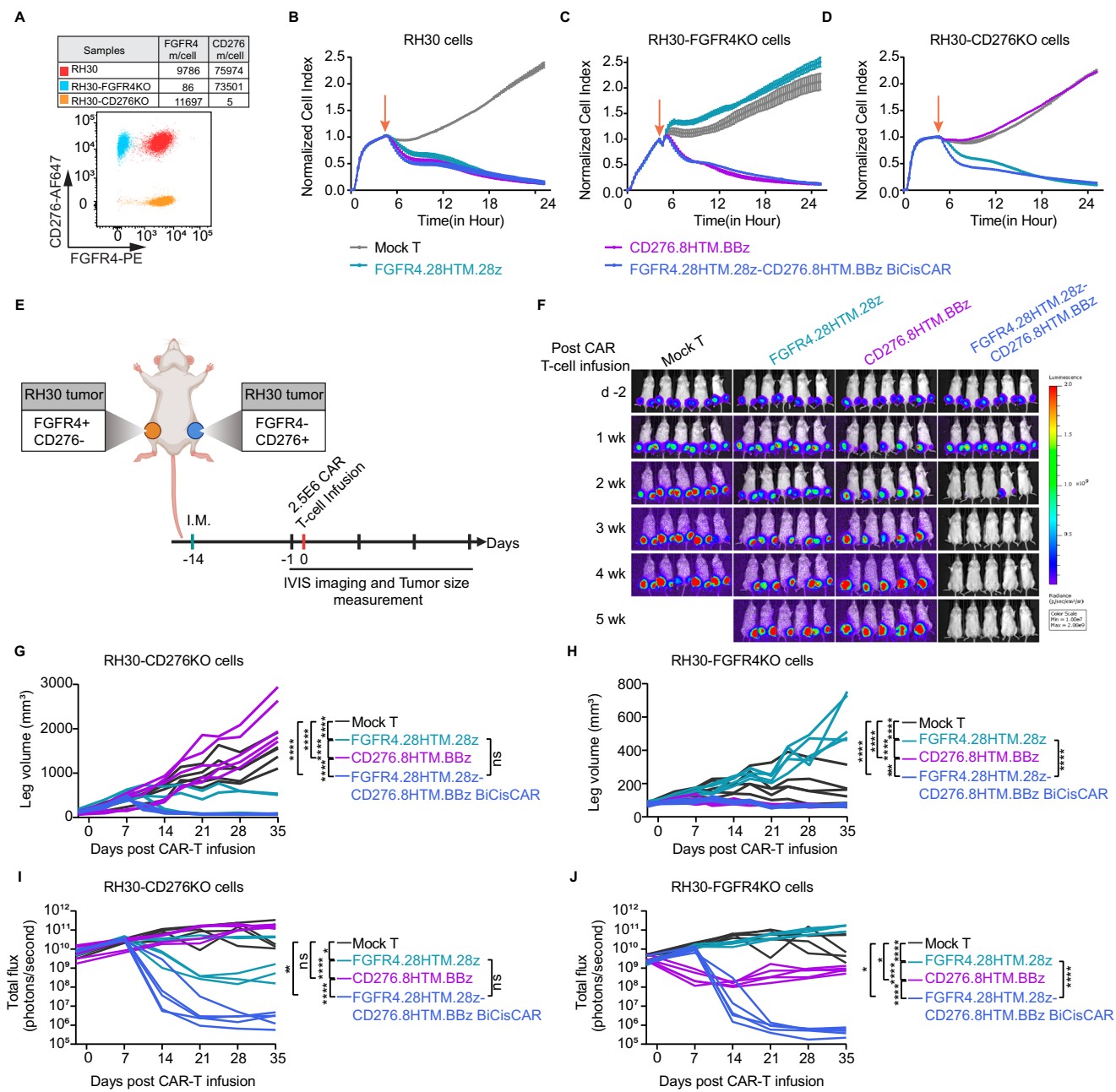

**Fig. 6 | BiCisCAR T-cells overcome heterogeneous expression of FGFR4 and CD276 in vivo. A** A representative flow-cytometric plot demonstrating the surface expression of FGFR4 and CD276 in RH30 (red), RH30-FGFR4KO (blue), or RH30-CD276KO (orange) cells. The top table shows the means of FGFR4 or CD276 molecules per cell. **B–D** Cytotoxicity assays using an xCELLigence RTCA show FGFR4.28HTM.28z-CD276.8HTM.BBz BiCisCAR T-cells continued to effectively kill FGFR4KO or CD276KO RH30 cells at an E:T ratio of 1:1. In contrast, FGFR4.28HTM.28z CAR or CD276.8HTM.BBz CAR did not induce cytolysis to FGFR4-KO or CD276KO RH30 cells respectively at an E:T ratio of 1:1. The orange vertical arrows indicate the time point at which CAR T-cells were added to the plate seeded with the target cells. **E** Schema of the heterogeneous RH30 I.M model infused with 2.5E6 CAR T-cells on day 14 following RH30-FGFR4KO (right) or RH30-CD276KO (left) tumor inoculation (created with BioRender.com). **F** Representative bioluminescence images of RH30-FGFR4KO or RH30-CD276KO cell growth in the I.M. model before and after CAR T-cell treatment. **G**, **H** Tumor size was monitored over 35 days by measuring leg volume before and after receiving mock or CAR T-cell treatment. Each replicate per group ($n = 5$) is shown. Two-way repeated measures (RM) ANOVA analysis was used to calculate the $p$-values between each paired group. ***$p = 0.0001$, ****$p < 0.0001$, by two-way RM ANOVA. Show bioluminescence kinetics of RH30-FGFR4KO (**I**) or RH30-CD276KO (**J**) following CAR T-cell treatment, using total flux values (photons per second). Data is presented as means ± SEM, $n = 5$. *$p = 0.0352$, **$p = 0.0099$, ****$p < 0.0001$ in **I**; *$p = 0.0278$ for Mock T versus CD276.8HTM.BBz, *$p = 0.014$ for Mock T versus FGFR4.28HTM.28z-CD276.8HTM.BBz BiCisCAR, and ****$p < 0.0001$ in **J**, as determined by two-way RM ANOVA. Source data is provided as a Source Data file.

(Clone O323), CD95 (Clone DX2), CD62L(Clone DREG-56), CD25 (Clone BC96), CD137 (Clone 4B4-1), LAG-3 (Clone 11C3C65), CD39 (Clone A1), PD-1 (Clone EH12.2H7), TIM-3 (Clone F38-2E2), Mouse IgG1, κ isotype (Clone MOPC-21), Mouse IgG2a, κ isotype (Clone MOPC-173), Mouse IgG2b, κ isotype (Clone MPC-11). After 3 washes, TILs were resuspended in PBS and counted, and then replicates in each group were combined before proceeding immediately to single-cell immune profiling using a Chromium Single Cell 5′ Solution v2 platform system (10× Genomics). 10× Genomics 5′ single-cell gene expression and cell-surface protein tag (CITE-seq) libraries were prepared as instructed by the 10× Genomics user guides. Libraries were sequenced on an Illumina NextSeq500 sequencer.

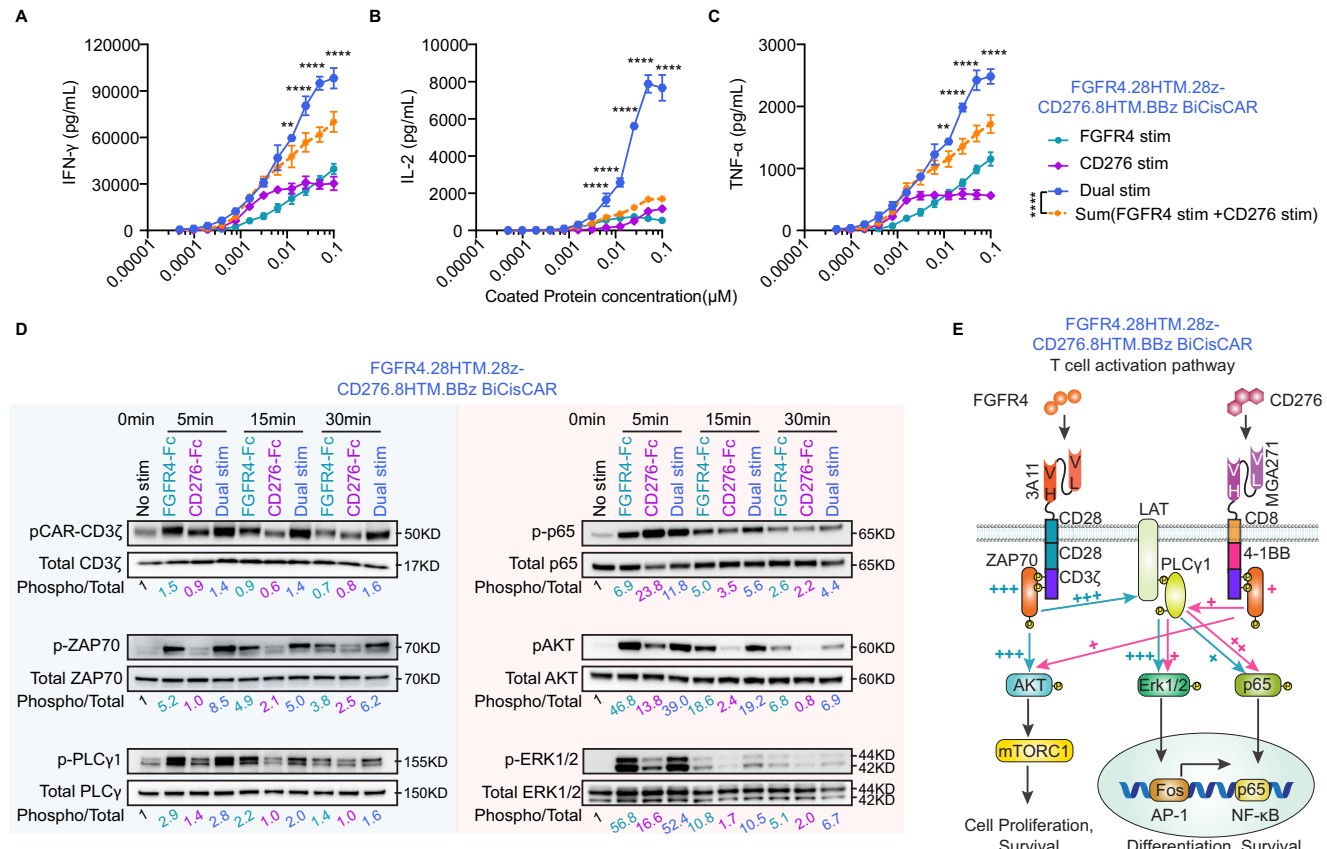

**Fig. 7 | Presence of both CD28 and 4-1BB CSDs in FGFR4 and CD276 dual-targeting CAR T-cells leads to heightened and sustained T-cell activation signaling.** IFN-γ (**A**), IL-2 (**B**), and TNF-α (**C**) were released by FGFR4.28HTM.28z-CD276.8HTM.BBz BiCisCAR T-cells following a 20-h stimulation with plate-coated FGFR4-Fc, CD276-Fc, or both proteins. Data are shown as the mean ± SD for 3 independent experiments. The orange dotted line shows the sum of cytokine release after a single protein stimulation. Two-way ANOVA Sidak's multiple comparisons test was performed for the difference between two proteins dual stimulation and the sum of single stimulation. **$p = 0.0069 in **A**, **$p = 0.0052 in **C**, ****$p < 0.0001$; ns not significant. **D** Western blot analyses of CAR-CD3ζ, ZAP70,

PLCγ1, p65, Akt and Erk1/2 phosphorylation in FGFR4.28HTM.28z-CD276.8HTM.BBz BiCisCAR T-cells after CAR stimulation (FGFR4-Fc protein for FGFR4 CAR, CD276-Fc protein for B7-H3, or both proteins for dual CAR activation) in a time course experiment. Numbers under the gel images indicate the ratio of signal intensity obtained with phospho-specific antibodies relative to that of the total protein. Relative values were normalized to one of the unstimulated controls. These results are representative of three independent experiments conducted with distinct T-cell donors. **E** Summary diagram outlining the activation signaling pathway of FGFR4.28HTM.28z-CD276.8HTM.BBz BiCisCAR T-cells following dual stimulation with FGFR4 and CD276. Source data is provided as a Source Data file.

Reads were aligned with human reference genome version GRCh38 and CAR binder scFv region sequences and quantified with Cell Ranger 7.0.1 (10× Genomics) using the standard workflow, which allowed for gating on CAR⁺ T-cells. The Cell Ranger raw output was imported into R (v4.2.1) using the Seurat package (v 4.3.0). The following filters were applied using the subset function to select high-quality live human single CAR T-cells: nFeature_RNA > 400 & nFeature_RNA < 8000; percent mitochondrial reads <20%. Single-cell protein data (representing the quantification of antibody-derived tags (ADT) in CITEseq data) was normalized by the DSB method as previously reported[17,56], which removes technical noise associated with unbound antibodies. The data matrices from 5 experimental groups, a total of 10 samples, were merged and then scaled and transformed with the SCTransform pipeline[57] to remove batch effect among different lanes. Weighted nearest neighbors (WNN) analysis of both merged RNA and ADT modalities was performed to obtain a UMAP reduction based on the above WNN graph, and a total of 21 clusters were generated with resolution = 0.4[58]. Clusters were defined by common markers analysis (Seurat FindAllMarkers with min.pct = 0.25 and logfc.threshold = 0.25; Supplementary Data 1) and protein markers expression (Supplementary Fig. 10). Cluster 12, 13, 15, and 19 of very few cells were combined into cluster 0 and annotated as CD4⁺ Activated T-cells, because they expressed similar RNA and protein profiling

Despite their different TCR clone sequences. Similarly, cluster 16 and 17 were merged into cluster 1 and defined as CD8+ Early Tem (GZMK⁺CD27⁺). Cells in cluster 4 exhibited high expression of mitochondria genes, while cells in cluster 8 had low nCounts, nFeatures and high percentage of mitochondria genes. Therefore, both cluster 4 and 8 were removed from the analysis due to their low quality.

A differential gene expression assay was performed between FGFR4.28HTM.28z-CD276.8HTM.BBz BiCisCAR and the other four CAR groups to identify the genes listed specifically and highly expressed on the TILs of FGFR4.28HTM.28z-CD276.8HTM.BBz BiCisCAR group. Data matrix was first normalized using a SCTransform (SCT) assay. Then differentially expressed genes were identified by the 'FindMarkers' function in Seurat with parameter logfc.threshold = 0 using the non-parametric Wilcoxon rank sum test.

**Cytokine production stimulated by FGFR4 or CD276 proteins**

Serial diluted FGFR4-Fc or CD276-Fc chimera protein were first coated on 96-well flat bottom cell-culture plates overnight at 4 °C. After washed once with PBS, 1E + 4 CAR⁺ T-cells were plated in 200 μl AIM-V medium (ThermoFisher, Cat# 12055091) per well for 20 h. Activation of CAR T cell was performed in triplicates for each experimental condition. Culture supernatants were collected and measured for the

production of IFN-γ, IL-2, and TNF-α by V-PLEX Custom Human Biomarkers Proinflammatory Panel 1 kit.

## Activation of CAR T-cells for western-blotting assay

To assess signaling through the CAR, 2E + 6 of BiCisCAR T-cells after starving for 4 h were placed on ice for 5 min, and then 1 μg/ml of human FGFR4-Fc Chimera Protein (Abcam, Cat# ab83999) 1 μg/ml of Human B7-H3 (4Ig)-Fc Chimera Protein (Acro Biosystem, Cat# B73-H82F5) were added and incubated for 30 min on the ice. Subsequently, cross-linking was induced by adding 5 μg/ml of AF647 conjugated Fab specific for human IgG-Fc (Jackson ImmunoResearch Laboratories, Cat# 109-607-008) to the indicated tubes and incubated at 37 °C for the indicated periods. At the end of the period of stimulation, cells were quenched on ice for 5 min and spun at 4 °C, then cell pellets were resuspended in 100 μL of Pierce RIPA buffer (Thermo Fisher Scientific, Cat# 89900) supplemented with Halt protease/phosphatase inhibitors (Thermo Fisher Scientific, Cat# 78442).

Western blot analysis was performed as previously described[7,28]. Briefly, cells were lysed by sonication, incubated on ice for 30 minutes, and then lysates were centrifuged for 20 min, the supernatant removed, and protein concentration quantified using a BCA (Promega) assay. 20 μg of lysate protein was resolved on 4–12% Bis-Tris gels (Invitrogen, Cat# NP0336) and transferred to PVDF membrane, blocked in 5% nonfat milk in Tris-buffered saline and Tween-20 (TBS-T). Membranes were incubated at 4 °C overnight in the following primary antibodies purchased from Cell Signaling Technology: anti-phospho-CD3ζ (Y142) (Clone EP265(2)Y, Cat# 1b68235, Abcam, 1:1000), anti-phospho-Zap-70 (Tyr319)/Syk (Tyr352) (Clone 65E4, Cat #2717, 1:1000), anti-Zap-70 (Clone L1E5, Cat #2709, 1:1000), anti-Phospho-PLCγ1 (Tyr783) (Clone D6M9S, Cat #14008, 1:1000), anti-PLCγ1 (Clone D9H10, Cat#5690, 1:1000), anti-phospho-ERK1/2 (T202/Y204) (Clone D13.14.4E, Cat# 4370S, 1:2000), anti-ERK1/2 (Cat# 9102S, 1:2000); anti-phospho-Akt (Ser473) (Clone D9E, Cat# 4060S, 1:2000) and anti-Akt (pan) (Clone C67E7, Cat# 4691S, 1:2000), anti-phospho-NF-κB p65 (Ser536) (Clone 93H1, Cat# 3033, 1:1000), anti-NF-κB p65 (Clone D14E12, Cat# 8242, 1:1000). Anti-CD3ζ (Clone 6C10.2, Cat# SC-1239, 1:2000) was purchased from Santa Cruz Biotechnology. After 3 times washing in TBS-T, membranes were then incubated with HRP-conjugated goat anti-mouse or goat anti-rabbit IgG (both Santa Cruz) at a dilution of 1 to 5000 and developed with ECL Prime (Amersham, Cat# RPN2232) on the gel station (Bio-Rad). Densitometric analysis of the phosphorylation-specific antibodies was performed using Image-Lab software. Phosphorylation levels were determined by calculating the ratio of the intensity of the signal obtained with phospho-specific antibodies relative to the total. Relative values were normalized to one of the untreated controls in every gel.

## Statistics

GraphPad Prism 8 software was used for graphing and data analysis. Log-rank statistical tests were used for survival analyses. An unpaired, two-tailed Student's *t* test was used to calculate the significant difference between 2 groups. An ordinary one-way ANOVA was used for multiple group comparisons. A two-way ANOVA was used for in vitro cytokine production data, in vivo tumor growth curves, and bioluminescence signal analysis. *p*-values of less than 0.05 were considered statistically significant. The statistical analysis method is also described in the individual figure legends.

## Reporting summary

Further information on research design is available in the Nature Portfolio Reporting Summary linked to this article.

## Data availability

Raw and processed data from the scRNA, and surface protein sequencing experiments are deposited in the NCBI's Gene Expression Omnibus (GEO, RRID: SCR_005012) database under accession code GSE228127. Publicly available ChIP-seq data used in this study are available in the GEO database under accession codes GSE83728 and GSE116344. The remaining data are available within the Article, Supplementary Information or Source Data files. Source data are provided with this paper.

## Code availability

Multimodal single-cell analysis code for CAR TILs from the JR IM model is available at: https://github.com/CCRGeneticsBranch/Khanlab_CITEseq_Tumor-Infiltrating-CART.

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

## Acknowledgements

This work was supported by the Intramural Research Program of the NIH, NCI, Center for Cancer Research (ZIA BC 010806, Developing Novel Therapies for Neuroblastoma and Rhabdomyosarcoma to J.K.) and an Enzo & Me Pediatric Cancer Foundation Grant for Rhabdomyosarcoma Research award (to M.T.) from Children's Oncology Group (COG)

Foundation. The authors acknowledge the members of NCI CCR Single Cell Analysis facility for assistance with multimodal single-cell assays. We thank Yanyu Wang of the Lymphokine Testing Section at the NCI Clinical Support Laboratory (CSL) for measuring cytokine production. We acknowledge the NCI Flow Core for providing instruments and assistance related to flow cytometry. Panel A in Fig. 5, panel E in Fig. 6, and Featured Figure are created with BioRender.com released under a Creative Commons Attribution-NonCommerical-NoDerivs 4.0 International license.

## Author contributions

J.K. and M.T. supervised the study and acquired funding. M.T. and J.S.W. designed most experiments, and S.B. and Z.Z. designed testing TCR activation signaling proteins experiment. M.T., A.T.C., D.M., Y.Y.K., Z.Z., E.G.P., A.R., C.L., and J.T.W. performed the experiments. M.T. collected, analyzed, and interpreted the data. M.C.K. provided technical support for the CITE-Seq experiment, and M.T., H.C.C., C.L., and X.W. analyzed multimodal single-cell assay data. J.S.W., D.M., Y.Y.K., J.T.W., S.B., Z.Z., and C.L. provided technical or material support. M.T. wrote the original draft of the manuscript. J.K., J.S.W., M.T., D.M., Y.Y.K., Z.Z., and S.B. reviewed and edited the manuscript. M.T., Y.Y.K., E.G.P., J.S.W., and J.K. revised the manuscript.

## Competing interests

J. K. and A.T.C. are inventors on international patent application no. PCT/US2016/052496. The 3A11 CAR sequence is in this patent application (see https://patents.justia.com/patent/11078286) filed on September 19, 2016, titled "Monoclonal antibodies specific for fibroblast growth factor receptor 4 (FGFR4) and methods of their use". J.K. and M.T. are inventors on international patent provisional application no. 63/634,330, titled "Chimeric Antigen Receptors Targeting FGFR4 and/or CD276 and Use Thereof for the Treatment of Cancer". The remaining authors have declared that no conflict of interest exists.
