## [Peer Review File · Nature Communications]

CAR T-cells targeting FGFR4 and CD276 simultaneously show potent antitumor effect against childhood rhabdomyosarcomaREVIEWER COMMENTS

Reviewer #1 (Remarks to the Author):

In the submitted manuscript, the authors describe the development of a bispecific CART approach for rhabdomyosarcoma (RMS) that consists of targeting FGR4 and CD276, two established targets for RMS. The authors compare several bispecific CART population and identify that co-expression of a FGR4.28z and a CD276.41BBz CAR in T cells endows CARTs with superior antitumor activity in several models. The authors also perform extensive correlative studies, demonstrating that the expression CARs with CD28 and 41BB costimulatory domains has additive and/or synergistic effects. While new novel mechanisms are explored, believe that the conducted study is very impactful and present a significant step forward in developing bispecific CART products for solid tumors such as RMS. The data is well presented and the manuscript is well written. I mainly have some minor concerns, which are summarized below.

Minor

- 1) Figure Panels 6G-J show that bispecific CARTs have superior anti-tumor activity than monospecific CARTs against single antigen positive tumors in vivo; the bioluminescence data (6I,J) is most likely explained by 'bleed through' of the bioluminescence signal of the contralateral, growing tumor. However, even when one reviews the leg volume measurements (6G,H), FGR4.28z CARTs have decreased antitumor activity in comparison to bispecific CARTs against CD276KO tumors (6G), which does not occur with FGR4KO cells and CD276-CARTs (6H); also, Mock Ts seem to have significant antitumor activity against FGR4KO cells. All these issues are not described and/or discussed, which I believe creates confusion on multiple levels.
- 2) As the authors pointed out, they cannot evaluate toxicity in their in vivo models. However, recommend comparing GMCSF and IL6 production of mono- and bi-specific CART population post antigen-specific activation; two cytokines that have been implicated in CART-associated cytokine release syndrome.
- 3) Fig 1H,M: Plotting Kaplan-Meier curves in which one group has a 'n=3' raises significant concerns regarding scientific rigor and reproducibility; either increase group size or remove.
- 4) Fig 7D: please provide the data for the other 2 donors as supplementary data.
- 5) Fig 10: please provide exemplary flow plots in the appendix.
- 6) The authors provide a very nice phenotypic analysis of the different FGR4-CART populations (S2C-F). However, such an analysis is not provided for the bispecific CART population (S5A) – believe that it would be very reassuring if a similar analysis is provided for these constructs. In particular, since one of the bispecific CART population is identified 'as the winner' in subsequent experiments.
- 7) Lines 204-207: the authors states that they generated bicistronic CARs; this is technically incorrect: the CAR is not bicistronic - they generated a bicistronic lentiviral vector encoding 2 CARs separated by a self-cleaving peptide – please correct.
- 8) Line 257: do not believe that these tumors are 'large' based on the provided Figure panel – recommend removing.
- 9) While the authors cite Hirabayashi et al (ref 28), they do not discuss that these authors published a different bispecific CART design (same TM), only one zeta domain – believe that this deserved to be discussed.

Reviewer #2 (Remarks to the Author):

Tian et al. developed a new bicistronic CAR T cell therapy to treat rhabdomyosarcoma by targeting FGFR4 and CD276. The authors mainly demonstrate that:

1. CD276 can be a novel antigen of rhabdomyosarcoma for CAR-T therapy
2. Bicistronic CAR T cell with CD28 and 4-1BB co-stimulatory signal synergizes CAR-T activation signal, leading to enhancement of overall CAR-T anti-tumor response.

Overall, this paper is well-structured, and the logic is easy to follow. However, some issues need to be addressed.

Comments

1. In Fig 1. the authors claim that substituting HTM from CD8 to CD28 can enhance tumor killing by demonstrating FGFR4-28HTM-BBz showed superior anti-tumor activity in vivo (Fig. 1E). However, it is quite interesting that no significant difference in tumor killing in vitro and cytokine produce between FGFR4-28HTM-BBz. Please provide potential reasons that could explain the difference in vivo and in vitro data.

2. In Fig 1N, there was a 100-fold higher number of FGFR4-28HTM-BBz detected in mice blood compared to FGFR4-28HTM-28z. Typically, better CAR-T expansion shows better tumor control, but authors found that FGFR4-28HTM-28z with lower number of CAR-T in mice blood has better tumor control activity compared to FGFR4-28HTM-BBz. Although the authors claim that CD28-based CAR-T showed faster killing but was easily exhausted, I think it is still quite challenging to understand that less tumor control activity found in CAR-T, which can expand 100 times more. It would be appreciated if the author could explain this inconsistency between tumor killing and CAR-T expansion.

3. Authors identified that bicis CAR with FGFR4-28HTM-BBz and CD276-8HTM-BBz showed higher exhausted phenotype than CAR with one CD28 and one BB in co-stimulatory domain. Please explain why two BB co-stimulatory signals can cause more exhaustion on the T cells.

4. Like Fig 1, there is an inconsistency between tumor control and CAR-T expansion in mice blood (i.e., bicis CAR with FGFR4-28HTM-BBz and CD276-8HTM-BBz has almost no tumor killing in vivo in 4E, but second higher expansion capability found in mice blood (Fig 4F)). Please provide an explanation.

5. Have authors switched the co-stimulatory signals to optimize bicis CAR-T construct? If yes, please provide data (i.e., FGFR4-8HTM-BBz and CD276-28HTM-28z).

Reviewer #3 (Remarks to the Author):

The manuscript by Tian and colleagues reports on the effect of bicistronic CARs targeted against FGFR4 and CD276 in rhabdomyosarcoma. Optimized CARs containing two different co-stimulatory domains increased anti-tumour activity significantly. The bicistronic CAR approach has the potential to reduce the risk of tumour escape, but also allows for additive and even synergistic activation as two co-stimulatory domains could result in increased down-stream signalling.

The manuscript is well-written and experiments nicely presented. The animal experiment using individual CRISPR knockouts of the targets is a particularly elegant approach to show specificity of the targeting and increased activity of the dual CARs.

FGFR4 levels are claimed by the authors to be low in all normal tissues and are clearly high in RMS tumours. However, FGFR4 has several important functions, including regulating bile acids, cholesterol and lipid metabolism. It is therefore a concern for side effects in children treated with a FGFR4-targeted CAR. The targeting of both FGFR4 and CD276 could also increase unwanted side effects.

Comments:

1. As the animal studies are performed in NSG mouse, it would be interesting to know if the CARs also recognize the mouse forms of FGFR4 and CD276? If they do, non-tumour targeting activities of the CARs could be analyzed in the animal models.
2. The authors suggest that the limited efficacy of the FGFR4 CAR in RMS559 may be due to heterogeneous FGFR4 expression (line 159). From the flow cytometry measurements in figure 2C, this does not seem to be the case. In any case, the hypothesis should be investigated more closely. Possibly, immunofluorescence microscopy could be used to investigate if FGFR4 is heterogeneously expressed on the cell-surface of RMS559. As FGFR4 is mutated in this cell line (FGFRV550L), the localization or stability of the receptor may be different.
3. It is surprising that the expression levels of FGFR4 and CD276 are so different in RMS559 compared to its xenograft counterpart RMS559 CDX (Fig 2D). This warrants an explanation.
4. It is claimed that dual stimulation leads to increased and more persistent downstream signalling compared to stimulation by FGFR4 and CD276 alone (Fig 7E). This is difficult to see in figure 7D, especially in the case of p-PLCgamma, pERK1/2 and pAKT. Actually, the dual stimulation signals seem very similar to FGFR4 stimulation alone. Western blotting is not very quantitative, and in this experiment, only one value for each condition is shown, which is not very convincing. Even if one representative western blot is shown, the quantification should be performed and analysed for all three independent experiments together. Then error bars and significant levels can be calculated.

REVIEWER COMMENTS

Reviewer #1 (Remarks to the Author):

In the submitted manuscript, the authors describe the development of a bispecific CART approach for rhabdomyosarcoma (RMS) that consists of targeting FGR4 and CD276, two established targets for RMS. The authors compare several bispecific CART population and identify that co-expression of a FGR4.28z and a CD276.41BBz CAR in T cells endows CARTs with superior antitumor activity in several models. The authors also perform extensive correlative studies, demonstrating that the expression CARs with CD28 and 41BB costimulatory domains has additive and/or synergistic effects. While new novel mechanisms are explored, believe that the conducted study is very impactful and present a significant step forward in developing bispecific CART products for solid tumors such as RMS. The data is well presented and the manuscript is well written. I mainly have some minor concerns, which are summarized below.

We thank the reviewer for the positive comments and the recognition that the work is "very impactful and presents a significant step forward in developing bispecific CART products for solid tumors such as RMS" and "perform extensive correlative studies". We have addressed all of the minor comments below, and clarified these in the revised manuscript.

Minor

1) Figure Panels 6G-J show that bispecific CARTs have superior anti-tumor activity than monospecific CARTs against single antigen positive tumors in vivo; the bioluminescence data (6I,J) is most likely explained by 'bleed through' of the bioluminescence signal of the contralateral, growing tumor. However, even when one reviews the leg volume measurements (6G,H), FGR4.28z CARTs have decreased antitumor activity in comparison to bispecific CARTs against CD276KO tumors (6G), which does not occur with FGR4KO cells and CD276-CARTs (6H); also, Mock Ts seem to have significant antitumor activity against FGR4KO cells. All these issues are not described and/or discussed, which I believe creates confusion on multiple levels.

Thank you for your careful review. We agree that although we attempted to gate carefully on the specific FGFR4KO and CD276KO tumor, the bioluminescence data (Fig. 6, I and J) reflect some 'bleed through' of signal from the contralateral tumor. Nevertheless, the results demonstrate that individual CAR T cells are effective against tumors with high expression of the target antigen, and the bispecific CAR T-cells are effective in eliminating both KO tumors.

We agree with the reviewer's observation concerning the decreased efficacy of FGFR4.28HTM.28z CAR T-cells compared to BiCisCAR against CD276KO tumor. We believe that this is due to its short persistence and proneness to exhaustion, shown in current

Fig. 1, L and M. On the contrary, BiCisCAR T-cells using two different co-stimulatory domains (CSDs) demonstrated better expansion and prolonged persistence in different RMS IM models (Fig. 4, H and J; Fig. S7G). We have added a description of this observation to clarify the results (lines 392-396).

Regarding the smaller tumor volume in the mock T-cells treated group compared with the CAR T-cells group, we believe this resulted from a significantly higher tumor burden in this cohort. We observed that these control mice had cachexia and were much frailer than those in CAR T-cell-treated groups. We believe the cachexia resulting from higher tumor burden was associated with slower growth of both CD276 KO (Fig. 6G) and FGFR4 KO (Fig. 6H) treated with mock T-cells. Moreover, this effect is further exacerbated by the fact that KO of FGFR4 leads to reduced growth of RMS tumors, as we have previously demonstrated [PMID: 19809159] and is readily demonstrated in comparing the tumor growth of the CD276 KO and FGFR4 KO cells compared with the parental cell line shown below in Rebuttal Figure 1.

Rebuttal Figure 1: Leg volume measured after tumors were intramuscularly implanted into NSG mice showed slower growth of RH30 tumors with FGFR4 KO.

2) As the authors pointed out, they cannot evaluate toxicity in their in vivo models. However, recommend comparing GMCSF and IL6 production of mono- and bi-specific CART population post antigen-specific activation; two cytokines that have been implicated in CART-associated cytokine release syndrome.

We thank the reviewer for their suggestion. In this study, we measured cytokine release of IFN- γ , IL-2, and TNF- α , but not specifically of GMCSF or IL-6. However, we expect they are all produced by CAR T-cells upon antigen-specific activation. Additionally, we did not observe any adverse effects, such as weight loss, in mice treated with BiCisCAR T cells from two independent donors (Rebuttal Figure 2). We do recognize this

potential side-effect associated with CAR T therapy, which will be closely monitored in future clinical trials.

Rebuttal Figure 2. Mice bearing RMS559 IM xenografts didn't lose weight after Mock T or BiCisCAR T-cells treatment. Each line represents the relative weight of a mouse before (0 day) and after CAR T-cell treatment from two T-cell donors.

3) Fig 1H,M: Plotting Kaplan-Meier curves in which one group has a 'n=3' raises significant concerns regarding scientific rigor and reproducibility; either increase group size or remove.

We appreciate the carefulness of the reviewer and agree to remove these two panels from the original Fig. 1. We also repeated these two experiments using CAR T-cells derived from another donor with an increased group size (6 mice per group) at the reviewer and the editor's request. The new experiment data are consistent with those in Fig. 1 and have been added in the new Fig. S3.

4) Fig 7D: please provide the data for the other 2 donors as supplementary data.

We have added the T-cell activation signaling data for the BiCisCAR T-cells from the other 2 donors to Fig. S13.

5) Fig 10: please provide exemplary flow plots in the appendix.

Because we simultaneously measured 4 exhaustion markers by flow cytometry, we used R to plot current Fig. 1M showing the expression of >2 markers on each same cell. As requested, representative flow plots for two markers for the FGFR4-targeting CAR T-cells are shown below (Rebuttal Figure 3). Additionally, the current flow software does not generate figures with >2 markers, and these are difficult to visualize.

Rebuttal Figure 3. Representative flow cytometry plots show the expression levels of CD39, PD-1, LAG-3, and Tim-3 on FGFR4 targeting CAR T-cells obtained from mouse blood after 32 days post-infusion into RH30 I.M. xenografts model.

6) The authors provide a very nice phenotypic analysis of the different FGR4-CART populations (S2C-F). However, such an analysis is not provided for the bispecific CART population (S5A) – believe that it would be very reassuring if a similar analysis is provided for these constructs. In particular, since one of the bispecific CART population is identified ‘as the winner’ in subsequent experiments.

Thanks for the reviewer’s comments and suggestions. We have now added a similar figure of phenotype analysis for three BiCisCAR T-cells in Fig. S6.

7) Lines 204-207: the authors states that they generated bicistronic CARs; this is technically incorrect: the CAR is not bicistronic - they generated a bicistronic lentiviral vector encoding 2 CARs separated by a self-cleaving peptide – please correct.

We agreed and modified the text accordingly. “we generated 3 bicistronic lentiviral constructs encoding 2 CARs to target both FGFR4 and CD276, separated by a self-cleaving linker allowing co-expression of both CARs in the same T-cell.”

8) Line 257: do not believe that these tumors are ‘large’ based on the provided Figure panel – recommend removing.

We agree with this comment and have removed the word “large” here.

9) While the authors cite Hirabayashi et al (ref 28), they do not discuss that these authors published a different bispecific CART design (same TM), only one zeta domain – believe that this deserved to be discussed.

We have added a discussion of the dual-targeting CAR design in this reference (lines 521-526).

Reviewer #2 (Remarks to the Author):

Tian et al. developed a new bicistronic CAR T cell therapy to treat rhabdomyosarcoma by targeting FGFR4 and CD276. The authors mainly demonstrate that:

1. CD276 can be a novel antigen of rhabdomyosarcoma for CAR-T therapy
2. Bicistronic CAR T cell with CD28 and 4-1BB co-stimulatory signal synergizes CAR-T activation signal, leading to enhancement of overall CAR-T anti-tumor response.

Overall, this paper is well-structured, and the logic is easy to follow. However, some issues need to be addressed.

We thank the reviewer for recognizing the main advances of the manuscript. We have adjusted the manuscript according to the reviewer's recommendations, which has considerably improved it.

Comments

1. In Fig 1. the authors claim that substituting HTM from CD8 to CD28 can enhance tumor killing by demonstrating FGFR4-28HTM-BBz showed superior anti-tumor activity in vivo (Fig. 1E). However, it is quite interesting that no significant difference in tumor killing in vitro and cytokine produce between FGFR4-28HTM-BBz. Please provide potential reasons that could explain the difference in vivo and in vitro data.

We appreciate the keen observation. Although not statistically significant, we do observe a trend of the FGFR4.28HTM.BBz CAR causing increased IFN- γ production compared to FGFR4.8HTM.BBz, in short-term (3 days) in-vitro experiments (Fig. 1C). We speculate that this insignificant but consistent difference resulted in different efficacy in mouse models over a longer period (6 weeks) (Fig. 1E). We confirmed this observation, and added FGFR4.28HTM.BBz *in-vivo* efficacy results in a new Fig. S1. Panels A to E in this new figure shows that the substitution with CD28 HTM does indeed enhance the antitumor activity of FGFR4.8HTM.BBz CAR in a low burden RH30 intramuscular xenograft model. A previous study reported that CD28 HTM significantly increased the recruitment of ZAP70 in CAR T-cells compared with CD8 HTM, especially at a low antigen density; moreover, CD28 HTM enhanced receptor clustering during T-cell

activation [PMID: 32193224]. These observations indicate that CD28 HTM imparts the CAR with greater potency. Additionally, by comparing the CAR T-cell counts circulating in the blood, we found FGFR4.28HTM.BBz CAR T-cells have higher persistence than those with CD8 HTM on day 31 after CAR T-cell infusion (Fig. S1F). We, therefore, confirmed that the CD28 HTM domain enhances the activity and persistence of the CAR T cells in vivo. We have added this in the conclusion to our manuscript in lines 124-125.

2. In Fig 1N, there was a 100-fold higher number of FGFR4-28HTM-BBz detected in mice blood compared to FGFR4-28HTM-28z. Typically, better CAR-T expansion shows better tumor control, but authors found that FGFR4-28HTM-28z with lower number of CAR-T in mice blood has better tumor control activity compared to FGFR4-28HTM-BBz. Although the authors claim that CD28-based CAR-T showed faster killing but was easily exhausted, I think it is still quite challenging to understand that less tumor control activity found in CAR-T, which can expand 100 times more. It would be appreciated if the author could explain this insistency between tumor killing and CAR-T expansion.

We believe there may have been some misunderstanding by the reviewer, for which we apologize. The current Fig. 1L (original Fig. 1N) refers to the RH30 model, in which two modified FGFR4-CARs showed similar high anti-tumor activity (Fig. 1, E to G). For the orthotopic RMS559 models, FGFR4-28HTM-BBz showed no difference in its persistence compared to FGFR4-28HTM-28z, except on day 32 (Former Fig S4G, new Fig S5G). We agree with the reviewer that a better CAR-T expansion is generally associated with better tumor control. So FGFR4.28HTM.BBz CAR indeed controls the growth of orthotopic xenografts in the RH30 model, the same as FGFR4.28HTM.28z CAR (Fig. 1, E to G). Even though this CAR showed significantly weaker killing activity and lower cytokine releases than that of FGFR4.28HTM.28z CAR (Fig. 1, B and C), the prolonged persistence and less exhaustion make this CAR functional for a long time, resulting in greater control of RH30 xenograft growth.

However, in the more aggressive, fast growing tumor, RMS559 model, rapid and greater anti-tumor activity endowed by the CD28 co-stimulatory domain becomes much more important in controlling and killing the tumor within the first 14 days (Fig. 1, I to K) before FGFR4.28HTM.28z CAR T-cells diminished (Fig. S5G). Although FGFR4.28HTM.BBz CAR showed modestly higher persistence on day 32 post-infusion (Fig. S5G), it couldn't control the growth of these aggressive RMS559 xenografts due to its weaker anti-tumor activity. Therefore, FGFR4-28HTM-28z is better at controlling rapidly

growing tumors compared to FGFR4-28HTM-BBz. We have conveyed this idea in lines 126-129 of our manuscript.

3. Authors identified that bicis CAR with FGFR4-28HTM-BBz and CD276-8HTM-BBz showed higher exhausted phenotype than CAR with one CD28 and one BB in co-stimulatory domain. Please explain why two BB co-stimulatory signals can cause more exhaustion on the T cells.

We thank the reviewer for raising this important question. We compared TCR activation signaling between two BiCisCARs following co-responding antigen stimulation in Figure S12D. BiCisCAR using the two 4-1BB CSDs induced lower phosphorylation levels of CAR-CD3z, PLC γ 1, AKT, and ERK, compared with BiCisCAR using CD28 and one BB in co-stimulatory domains after dual-stimulation, suggesting a weaker activation of BiCisCAR T-cells using two 4-1BB CSDs. Prolonged exposure to high levels of antigen is a major cause of T cell exhaustion (PMID: 26205583 and PMID: 21555851). Therefore, we believe prolonged exposure to antigen stimulation at the tumor site is likely the cause of more exhaustion for the BiCisCAR T-cells with two 4-1BB co-stimulatory domains.

4. Like Fig 1, there is an inconsistency between tumor control and CAR-T expansion in mice blood (i.e., bicis CAR with FGFR4-28HTM-BBz and CD276-8HTM-BBz has almost no tumor killing in vivo in 4E, but second higher expansion capability found in mice blood (Fig 4F)). Please provide an explanation.

Tumor control by the CAR T-cells is not only dependent on counts in blood at day 21 but also on rapid expansion and cytolytic activity of CAR T cells at an earlier time point to control rapidly growing tumors. We attempted to detect CAR T cells in blood circulation before 11 days without much success due to few CAR T cells detected by this flow cytometry method.

Thus, although there was a comparable CAR T cell count at day 21 for both BiCisCARs, the tumor cytotoxicity for FGFR4.28HTM.28z-CD276.8HTM.BBz CAR T cells were superior to that of FGFR4.28HTM.BBz-CD276.8HTM.BBz CAR T cells (Fig. 3F). We have a sentence in the discussion to reflect this in lines 506-509.

5. Have authors switched the co-stimulatory signals to optimize bicis CAR-T construct? If yes, please provide data (i.e., FGFR4-8HTM-BBz and CD276-28HTM-28z).

Thanks for raising this interesting question. We didn't switch the co-stimulatory signals in BiCisCARs, and it would be an interesting experiment. This important question is the

subject of a future study.

Reviewer #3 (Remarks to the Author):

The manuscript by Tian and colleagues reports on the effect of bicistronic CARs targeted against FGFR4 and CD276 in rhabdomyosarcoma. Optimized CARs containing two different co-stimulatory domains increased anti-tumour activity significantly. The bicistronic CAR approach has the potential to reduce the risk of tumour escape, but also allows for additive and even synergistic activation as two co-stimulatory domains could result in increased down-stream signalling.

The manuscript is well-written and experiments nicely presented. The animal experiment using individual CRISPR knockouts of the targets is a particularly elegant approach to show specificity of the targeting and increased activity of the dual CARs.

FGFR4 levels are claimed by the authors to be low in all normal tissues and are clearly high in RMS tumours. However, FGFR4 has several important functions, including regulating bile acids, cholesterol and lipid metabolism. It is therefore a concern for side effects in children treated with a FGFR4-targeted CAR. The targeting of both FGFR4 and CD276 could also increase unwanted side effects.

We thank the reviewer for the enthusiasm and comments that the manuscript was well-written, and experiments nicely presented and for the recognition of several novel aspects of the manuscripts.

For the reviewer's concern about targeting FGFR4, we have evaluated the toxicity of FGFR4.8HTM.BBz CAR T-cells using a panel of human primary cells. Co-culturing of these CAR T-cells with primary cells expressing low or absent FGFR4 cannot activate cytokine release by CAR T-cells, suggesting FGFR4 targeting CARs are safe (PMID: 37774704). Moreover, an FDA-IND application for using this CAR to treat FGFR4-expressing rhabdomyosarcoma has been recently approved by the FDA, to conduct a clinical trial. Thus, we will monitor the safety of FGFR4-targeting CAR T-cells further in a phase I clinical trial.

So far, two clinical trials (NCT04185038 and NCT04483778) have reported the safety of CD276-targeting CAR in treating patients with Diffuse intrinsic pontine glioma (DIPG) or children and young adults (CYA) with relapsed or refractory solid tumors (PMID: 36259971). Therefore, these existing data suggest CAR T-cell targeting of both FGFR4 and CD276 in humans is likely to be safe. However, we acknowledge that dual-targeting FGFR4 and CD276 may increase the risk of CAR therapy-related cytotoxic side effects, which requires us to evaluate in future clinical trials.

Comments:

1. As the animal studies are performed in NSG mouse, it would be interesting to know if the CARs also

recognize the mouse forms of FGFR4 and CD276? If they do, non-tumour targeting activities of the CARs could be analyzed in the animal models.

We agree that this is an important question, however, we could not test the non-tumor targeting activities in murine models because both binders of FGFR4 and CD276-targeting CAR only recognize human antigens (PMID: 37774704 and PMID: 22615450).

2. The authors suggest that the limited efficacy of the FGFR4 CAR in RMS559 may be due to heterogeneous FGFR4 expression (line 159). From the flow cytometry measurements in figure 2C, this does not seem to be the case. In any case, the hypothesis should be investigated more closely. Possibly, immunofluorescence microscopy could be used to investigate if FGFR4 is heterogeneously expressed on the cell-surface of RMS559. As FGFR4 is mutated in this cell line (FGFRV550L), the localization or stability of the receptor may be different.

We thank the reviewer for raising this interesting question. We plotted flow cytometry data for FGFR4 expression levels on the cell surface of RMS559 cells and 3 cell line-derived xenografts (CDX) (Rebuttal Figure 4). FGFR4 expression is indeed heterogeneous, spanning 2 log intervals on RMS559 cells (labeled with GFP), with a small proportion of cells with very low or absent expression of FGFR4. Thus, we believe the heterogeneous expression of FGFR4 may contribute to the poor efficacy of FGFR4 CAR.

Rebuttal Figure 4. Flow cytometry was used to measure FGFR4 expression level on RMS559 cell or cell line-derived xenografts (CDX) growing 29 days in NSG mice, by staining with mouse anti-human FGFR4 antibody (3A11).

3. It is surprising that the expression levels of FGFR4 and CD276 are so different in RMS559 compared to its xenograft counterpart RMS559 CDX (Fig 2D). This warrants an explanation.

We appreciate the carefulness of the reviewer. We were also surprised by the difference in antigen expression between RMS559 cell line and CDX. We, therefore, repeated this measurement using both flow cytometry and western blot (Rebuttal Figure 5). RMS559 CDXs consistently express lower cell surface FGFR4 and CD276 in single-cell suspension dissociated from CDXs than those on cells from cell culture. Furthermore, the western blot assay confirms the total FGFR4 expression is lower in RMS559 CDXs compared to cell lines (Rebuttal Figure 5B).

It is not clear what causes the difference in expression levels between cell lines and CDX, but our results are consistent. We surmise this is due to differences in local cytokines and growth in 3D of RMS559 in-vivo as CDXs versus in-vitro in cell line culture. We have added this as an observation in lines 185-186.

Rebuttal Figure 5. RMS559 CDXs express lower FGFR4 and CD276 than those in cells cultured in medium. (A) A representative flow cytometry plot shows the variation of FGFR4 or CD276 expression on RMS559 cell line and CDXs. The mean fluorescence intensity (MFI) of FGFR4 or CD276 on indicated samples is listed in the right table. (B) Total FGFR4 or CD276 protein levels expressed in RRMS559 cell and CDXs are determined by Western blotting analysis.

4. It is claimed that dual stimulation leads to increased and more persistent downstream signalling compared to stimulation by FGFR4 and CD276 alone (Fig 7E). This is difficult to see in figure 7D, especially in the case of p-PLCgamma, pERK1/2 and pAKT. Actually, the dual stimulation signals seem very similar to FGFR4 stimulation alone. Western blotting is not very quantitative, and in this experiment, only one value for each condition is shown, which is not very convincing. Even if one representative western blot is shown, the quantification should be performed and analyzed for all three independent experiments together. Then error bars and significant levels can be calculated.

We thank the reviewer's suggestion and have repeated this experiment using BiCisCAR T-cells from another 2 donors (Fig. S13). We also quantify the Western blots by including a dot graph to present phosphorylation changes of these TCR activation signaling proteins from 3 donors. Due to the large variation from 3 donors, we didn't see a significant difference between FGFR4 alone and dual antigens stimulation. However, we did observe a trend of higher levels of signaling protein activation in CAR-CD3z, ZAP70, P65, and AKT following dual stimulation than FGFR4-alone stimulation.

We modified the manuscript according to these changes. We thank the reviewers and believe their suggestions improved our manuscript significantly.

REVIEWERS' COMMENTS

Reviewer #1 (Remarks to the Author):

I would like to thank the authors for thoughtfully addressing all my concerns with this resubmission of the manuscript.

Reviewer #2 (Remarks to the Author):

Thank the authors for their kind response to the comments. all issues are resolved. There is no further question.

Reviewer #3 (Remarks to the Author):

The referee thanks the authors for answering all the comments. All is satisfactory answered, except the last question. It is understandably a lot of variation when three different donors are analysed, but in any case, the data provided are not convincing enough to claim a stronger and sustained effect of dual stimulation on down-stream signalling.

I suggest the authors tone down the conclusions of the signalling analyses as the data do not support the strong arguments, especially in the abstract, suggesting synergistic effects of dual stimulation. The statement "...through synergistic AKT, ERK1/2, and p65 signaling" is not supported by the data and should be changed.

Reviewer #1 (Remarks to the Author): I would like to thank the authors for thoughtfully addressing all my concerns with this resubmission of the manuscript.

Thank you.

Reviewer #2 (Remarks to the Author): Thank the authors for their kind response to the comments. all issues are resolved. There is no further question.

Thanks.

Reviewer #3 (Remarks to the Author): The referee thanks the authors for answering all the comments. All is satisfactory answered, except the last question. It is understandably a lot of variation when three different donors are analyzed, but in any case, the data provided are not convincing enough to claim a stronger and sustained effect of dual stimulation on down-stream signaling.

I suggest the authors tone down the conclusions of the signaling analyses as the data do not support the strong arguments, especially in the abstract, suggesting synergistic effects of dual stimulation. The statement "...through synergistic AKT, ERK1/2, and p65 signaling" is not supported by the data and should be changed.

We agree and have removed this statement from the abstract as suggested by the reviewer.